# SimpleFold: Folding Proteins is Simpler than You Think

**Yuyang Wang, Jiarui Lu***, **Navdeep Jaitly***, **Josh Susskind, Miguel Angel Bautista**
Apple

## Abstract

Protein folding models have achieved groundbreaking results typically via a combination of integrating domain knowledge into the architectural blocks and training pipelines. Nonetheless, given the success of generative models across different but related problems, it is natural to question whether these architectural designs are a necessary condition to build performant models. In this paper, we introduce *SimpleFold, the first flow-matching based protein folding model that solely uses general purpose transformer blocks*. Protein folding models typically employ computationally expensive modules involving triangular updates, explicit pair representations or multiple training objectives curated for this specific domain. Instead, SimpleFold employs standard transformer blocks with adaptive layers and is trained via a generative flow-matching objective with an additional structural term. We scale SimpleFold to 3B parameters and train it on approximately 9M distilled protein structures together with experimental PDB data. On standard folding benchmarks, SimpleFold-3B achieves competitive performance compared to state-of-the-art baselines, in addition SimpleFold demonstrates strong performance in ensemble prediction which is typically difficult for models trained via deterministic reconstruction objectives. SimpleFold challenges the reliance on complex domain-specific architectures designs in protein folding, opening up an alternative design space for future progress.

## 1 Introduction

Protein folding, the task of predicting a protein's three-dimensional atomic structure from its amino acid (AA) sequence, is a longstanding challenge in computational biology with far-reaching implications in drug discovery (Jumper et al., 2021; Baek et al., 2021). In this paper, we approach the protein folding problem purely from a generative modeling perspective without making strong assumptions about the natural generation process of protein structures. We draw parallels between protein folding and vision generative models (i.e., text-to-image or text-to-3D generation (Poole et al., 2022; Lin et al., 2023a; Hong et al., 2024a;b)), where the input AA sequence plays the role of a "text prompt" to a generative model which outputs the all-atom 3D coordinates. Inspired by the recent success of generative models in the vision domain we build a general-purpose yet powerful architecture based solely on standard transformer blocks with adaptive layers (Vaswani et al., 2017; Peebles & Xie, 2023) which we trained at larger scale than previous protein folding models, both in terms of model size and training data.

Established protein folding models like AlphaFold2 (Jumper et al., 2021) and RoseTTAFold (Baek et al., 2021) have achieved groundbreaking accuracy by relying on carefully engineered architectures that integrate computationally heavy domain-specific designs for protein folding tasks such as multiple sequence alignments (MSAs), pair representations, and triangle updates (Jumper et al., 2021; Baek et al., 2021). These design choices attempt to hard-code our current understanding of the underlying structure generation process into these models, instead of opting to let models to learn this directly from data, which could be beneficial for a variety of reasons. For example, (Lin et al., 2023b) showed that for orphan proteins (those with few or no close homologs) approaches based on protein language models (PLM) tend to outperform approaches that rely on MSA like AlphaFold2. Although folding models initially treated protein structure prediction as a deterministic problem via

---

*Work was completed while J.L. and N.J. were at Apple.

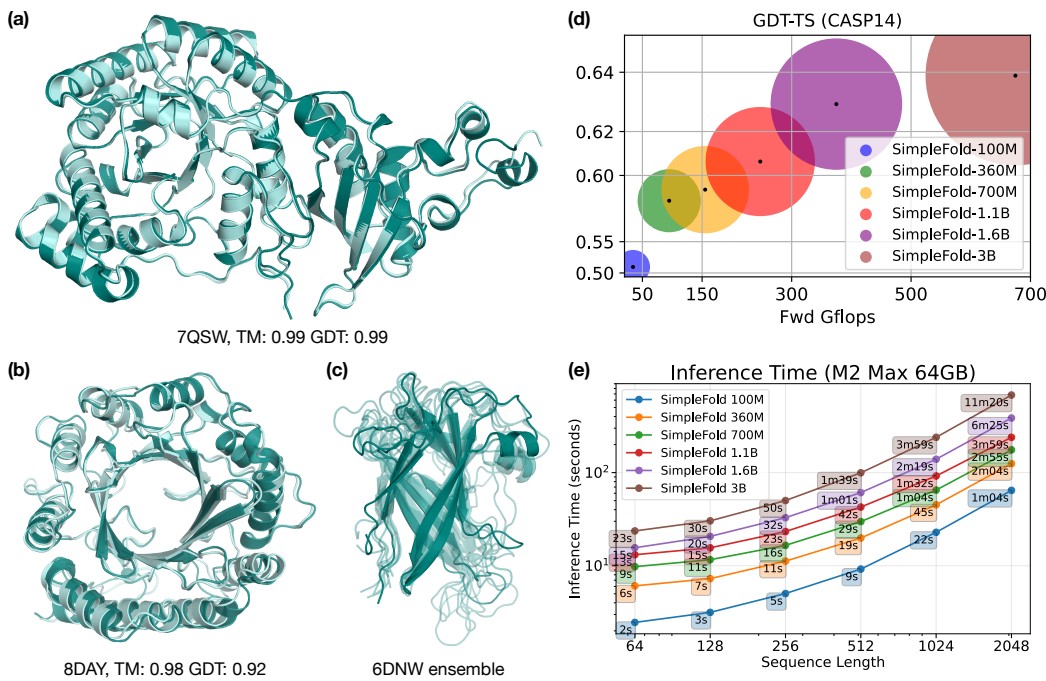

Figure 1: Example predictions of SimpleFold on targets (a) chain A of 7QSW (RubisCO large subunit) and (b) chain A of 8DAY (Dimethylallyltryptophan synthase 1), with ground truth shown in light aqua and prediction in deep teal. (c) Generated ensembles of target chain B of 6NDW (Flagellar hook protein FlgE) with SimpleFold finetuned on MD ensemble data. (d) Performance of SimpleFold on CASP14 with increasing model sizes from 100M to 3B. (e) Inference time of different sizes of SimpleFold on consumer level hardware, i.e., M2 Max 64GB Macbook Pro.

reconstruction objectives (Jumper et al., 2021; Baek et al., 2021; Lin et al., 2023b), recent works have explored building generative modeling for folding (Jing et al., 2024a). Generative approaches provide a way to model how native protein structures appear in nature, i.e., as a *non-deterministic* minimizer of the the Gibbs free energy of the atomic system. Generative models naturally capture this uncertainty and make it straightforward to generate ensembles of viable conformations instead of a single deterministic output (Jing et al., 2023; Abramson et al., 2024; Wohlwend et al., 2024; Bose et al., 2023; Watson et al., 2023b; Yim et al., 2023b;a; Geffner et al., 2025; Lin et al., 2024). However, these approaches still employ the expensive architectural components from AlphaFold2 like pair representations and triangle updates.

In this work, we propose *SimpleFold*, a flow-matching based folding model that directly maps a protein sequence to its full 3D atomic structure without relying on MSA, pair representations, triangular updates or any equivariant modules. Our architecture is inspired by recent transformer-based flow matching generative models (Peebles & Xie, 2023; Ma et al., 2024), with a strong emphasis on departing from current architecture designs using a general-purpose transformer backbone trained end-to-end with a flow-matching training objective. Crucially, we demonstrate that strong folding performance (see Fig. 1 can be achieved without explicit pairwise representations, triangle updates, or MSA, which significantly reduces architectural complexity and challenges preconceived notions around the necessity of these designs (Lin et al., 2023b). *SimpleFold* represents a strong departure from previous of protein folding models, and we summarize our contributions as follows:

- We revisit protein folding as a conditional generative task and introduce SimpleFold, a flow-based transformer folding model that eliminates MSA, pairwise representations, and triangle modules.

- We scale SimpleFold to 3B parameters and train it on approximately 9M distilled structures together with PDB experimental data.

- Our most powerful SimpleFold-3B shows strong results in folding compared to baselines with hard-coded heuristic designs and also achieves competitive performance on ensemble generation.

- We train a family of models ranging from 100M to largest 3B (see Fig. 1(d)). SimpleFold-100M recovers $\sim$90% performance of our best model on major folding benchmarks while being very efficient in inference even on consumer-level devices.

## 2 SIMPLEFOLD

### 2.1 FOLDING WITH FLOW-MATCHING

SimpleFold casts protein folding as a flow-matching generative model which generates protein structures from noise, conditioned on a given amino acid sequence. This "amino acid sequence-to-protein structure" generative model is conceptually very similar to "text-to-image" generative models. Flow-matching models (Lipman et al., 2023; Albergo & Vanden-Eijnden, 2023) approach generation as a time-dependent process that moves noise to data through integrating an ordinary differential equation (ODE) over time. For time $t \in [0, 1]$, flow matching defines a path of probability distributions $p_t(\mathbf{x}_t)$ that continuously transforms a tractable distribution $p_0$ (e.g., Gaussian) into an arbitrarily complex data distribution $p_\mathcal{D}$. In practice, the transformation is parameterized by a learnable time-dependent velocity field $\mathbf{v}_\theta(\mathbf{x}_t, t)$, and the generative process is defined by integrating the ODE, $\mathrm{d}\mathbf{x}_t = \mathbf{v}_\theta(\mathbf{x}_t, t) \, \mathrm{d}t$, from noise to data.

SimpleFold implements a linear interpolant path (Albergo & Vanden-Eijnden, 2023) (also referred to as a rectified flow (Liu et al., 2023; Esser et al., 2024)) between samples from the empirical data distribution $\mathbf{x} \sim p_\mathcal{D}$ and noise samples $\boldsymbol{\epsilon} \sim \mathcal{N}(\mathbf{0}, \boldsymbol{I})$, such that $\mathbf{x}_t = t\mathbf{x} + (1-t)\boldsymbol{\epsilon}$, where the target velocity is defined as $\mathbf{v}_t = \mathbf{x} - \boldsymbol{\epsilon}$. In flow matching, we train a network $\mathbf{v}_\theta$ to match the target across time and data via $\ell_2$ regression objective $\mathbb{E}[||\mathbf{v}_\theta(\mathbf{x}_t, t) - \mathbf{v}_t||^2]$.

In particular, given a protein with $N_a$ heavy atoms, we build a linear interpolant between noise $\boldsymbol{\epsilon}$ and all-atom positions $\mathbf{x}$, where $\boldsymbol{\epsilon}, \mathbf{x} \in \mathbb{R}^{N_a \times 3}$, conditioned on the amino acid sequence $\mathbf{s} \in \mathbb{R}^{N_r}$, where $N_r$ is number of residues or amino acids in the protein. Unlike earlier work that modeled only the $C_\alpha$ backbone with flow-matching models (Lin & AlQuraishi, 2023; Lin et al., 2024; Geffner et al., 2025), we generate full-atom conformations including both backbones and side chains.

**Training objective.** The network $\mathbf{v}_\theta$ takes the amino acid sequence as a conditioning input $\mathbf{v}_\theta(\mathbf{x}_t, \mathbf{s}, t)$ to model the target velocity field. In particular, the flow-matching objective is defined as follows:

$$\ell_{\text{FM}} = \mathbb{E}_{\mathbf{x},\mathbf{s},\boldsymbol{\epsilon},t} \left[ \frac{1}{N_a} \left\| \mathbf{v}_\theta(\mathbf{x}_t, \mathbf{s}, t) - (\mathbf{x} - \boldsymbol{\epsilon}) \right\|^2 \right]. \tag{1}$$

We also include an additional local distance difference test (LDDT) loss similar to (Abramson et al., 2024). This loss measures the atomic pairwise distances error between the generated structure $\hat{\mathbf{x}}(\mathbf{x}_t)$ at timestep $t$ and ground truth structures $\mathbf{x}$. During training, $\hat{\mathbf{x}}(\mathbf{x}_t)$ is estimated through one step Euler, i.e., $\hat{\mathbf{x}}(\mathbf{x}_t) = \mathbf{x}_t + (1-t)\mathbf{v}_\theta(\mathbf{x}_t, \mathbf{s}, t)$. The LDDT loss is formulated as follows:

$$\ell_{\text{LDDT}} = \mathbb{E}_{\mathbf{x},\mathbf{s},\boldsymbol{\epsilon},t} \left[ \frac{\sum_{i \neq j} \mathbb{1}(\delta_{ij} < \mathcal{C}) \sigma(\|\delta_{ij} - \hat{\delta}_{ij}^t\|)}{\sum \mathbb{1}(\delta_{ij} < \mathcal{C})} \right], \tag{2}$$

where $\delta_{ij} = \|\mathbf{x}_i - \mathbf{x}_j\|$ and $\hat{\delta}_{ij}^t = \|\hat{\mathbf{x}}(\mathbf{x}_t)_i - \hat{\mathbf{x}}(\mathbf{x}_t)_j\|$ denote the distances between atom $i, j$ in ground truth and predicted structures, respectively. The term $\sigma(\cdot)$ is a nonlinear function on pair distance errors and $\mathcal{C}$ is a cutoff distance which controls neighboring atoms to be included in the loss. The model is trained with a weighted combination of flow-matching and LDDT terms:

$$\ell = \ell_{\text{FM}} + \alpha(t)\ell_{\text{LDDT}}, \tag{3}$$

where $\alpha(t)$ is a weighting term related to timestep $t$ in flow process and is also dependent to different training phases (see Sect. 4.1).

**Timestep resampling.** To improve training efficiency and force generating structures with fine details (Esser et al., 2024; Geffner et al., 2025), the timestep $t$ is sampled from the distribution:

$p(t) = 0.98\text{LN}(0.8, 1.7) + 0.02\mathcal{U}(0, 1)$, where LN is logistic-normal distribution (Atchison & Shen, 1980) and $\mathcal{U}$ is a uniform distribution. Unlike popular timestep resampling in image generation (Esser et al., 2024), where timesteps are more densely sampled in the middle of the flow process (i.e., around $t = 0.5$), we shift the sample weight towards timesteps that are closer to clean data (i.e., $t = 1$), similar to findings in (Geffner et al., 2025) in the context of unconditional generation. This improves quality of generated samples especially in modeling refined structures of side chain atoms. We attribute this to the fact that protein structures contain a strong coarse-to-fine hierarchy "secondary structure - $C_\alpha$ backbone - side chain", thus oversampling close to the data manifold drives the model to better learn the refined atomic positions. Additional details regarding the LDDT loss and timestep resampling can be found in Appendix C.1.

## 2.2 ARCHITECTURE

Fig. 2 shows an architecture diagram of SimpleFold, which contains three major modules: light-weighted atom encoder and decoder which are symmetric (i.e., same number of blocks and hidden size) and a heavy residue trunk. All modules are implemented with standard transformer blocks with adaptive layers conditioned on the timestep $t$ (see bottom left of Fig. 2).

The atom encoder takes in "noisy" atomic coordinates $\mathbf{x}_t$ together with their corresponding atomic features (e.g., atomic type and charge, see Appendix A for details) and outputs atom tokens $a \in \mathbb{R}^{N_a \times d_a}$. In the atom encoder we use a local attention mask that constraints atom latents to only attend to a local neighborhood around their residue (i.e., atom tokens only attend to atom tokens of nearby residues in the sequence). The grouping operation takes the output of the atom encoder and conducts average pooling to atom tokens within the same residue to obtain residue tokens $\mathbf{r} \in \mathbb{R}^{N_r \times d_a}$ (see an illustration of grouping and ungrouping operations in Fig. 5).

Similar to text-to-image and text-to-3D generative models, we use a frozen pretrained protein language model (PLM) to embed the AA sequence into an informative latent representation. We leverage ESM2-3B (Lin et al., 2023b) in all our models to encode the AA sequence $\mathbf{s}$

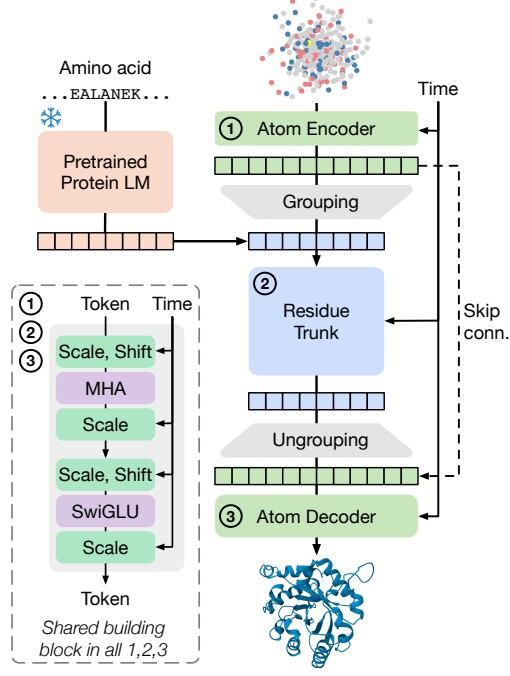

Figure 2: Overview of SimpleFold's architecture built on general-purpose standard Transformer block with adaptive layers. Atom encoder, residue trunk, and atom decoder all share the same general-purposed building block. Our model circumvents the need for pair representations or triangular updates.

into per-residue conditioning embeddings $\mathbf{e} \in \mathbb{R}^{N_r \times d_e}$. Sequence embeddings are then concatenated with the residue tokens along the channel dimension and fed into the residue trunk. The residue trunk contains most of the parameters of the model and is where most of the compute is spent on. The ungrouping operation projects residue tokens to corresponding atom tokens. Specifically, we broadcast the same residue token to the number of atoms a particular residue contains, which is defined by AA types. A skip connection from the output of atom encoder is also added to distinguish between different atoms within the same residue.

Finally, the atom decoder updates the atom tokens and outputs the predicted velocity field $\hat{\mathbf{v}}_t$. Local attention masks are also applied in the atom decoder as the encoder. The overall architecture of SimpleFold incorporates the hierarchical structure in proteins implementing a "fine - coarse - fine" scheme to balance the performance and efficiency.

SimpleFold strongly departs from the design choices in previous work (Chakravarty & Porter, 2022; Lin et al., 2023b; Abramson et al., 2024). Unlike AlphaFold2 (Chakravarty & Porter, 2022) or

ESMFold (Lin et al., 2023b) which explicitly keep a pair representation initialized by embeddings from expensive MSA search or attention score from the pretrained PLM, SimpleFold only keeps a single sequence representation which does not require triangle update and is thus far more efficient. In contraposition to previous works Lin et al. (2024); Chakravarty & Porter (2022); Lin et al. (2023b) which rely on equivariant architectures to generate physically meaningful results, SimpleFold is built on standard non-equivariant Transformer blocks.

## 2.3 SAMPLING

To fold a protein with a given amino acid sequence $\mathbf{s}$ in inference, we initialize atomic coordinates as Gaussian noise $\mathbf{x}_0 \sim \mathcal{N}(\mathbf{0}, \boldsymbol{I})$ and integrate the learned vector field from $t = 0$ to $t = 1$, which generates a full-atom structure corresponding to the input sequence. We perform stochastic generation using a Langevin-style SDE formulation of the flow process, leveraging the equivalence between the learned velocity field $\mathbf{v}_\theta$ and a score function $\mathbf{s}_\theta$, namely $\mathbf{s}_\theta(\mathbf{x}_t, \mathbf{s}, t) = (t\mathbf{v}_\theta(\mathbf{x}_t, \mathbf{s}, t) - \mathbf{x}_t)/(1 - t)$ (Albergo et al., 2023; Song et al., 2021). In particular, we apply the Euler–Maruyama integrator (Ma et al., 2024):

$$\mathrm{d}\mathbf{x}_t = \mathbf{v}_\theta(\mathbf{x}_t, \mathbf{s}, t)\,\mathrm{d}t + \frac{1}{2}w(t)\mathbf{s}_\theta(\mathbf{x}_t, t, c)\,\mathrm{d}t + \sqrt{\tau \cdot w(t)}\,\mathrm{d}\bar{\mathbf{W}}_t, \tag{4}$$

where $w(t) > 0$ is a time-dependent diffusion coefficient, $\bar{\mathbf{W}}_t$ is a reverse-time Wiener process, and $\tau$ controls the scale of stochasticity. We find $w(t) = \frac{2(1-t)}{t+\eta}$, which defines stochasticity scheduler following SNR of flow process and $\eta$ is a small constant for numerical stability, gives the best sampling quality. We stick to this setting in all our experiments unless mentioned otherwise. Similar to previous flow-matching based protein generative models (Geffner et al., 2025), we find that $\tau$ balances the generation of accurate refined structures and modeling the ensemble of conformations.

## 2.4 TRAINING ON DISTILLED DATA

We train SimpleFold with a data mix of 3 different sources. First, we include around 160K structures from PDB with a cutoff of May 2020 following ESMFold (Lin et al., 2023b). Additionally, we use the SwissProt set from AFDB. Within SwissProt distilled structures, we select samples with average pLDDT greater than 85 and standard deviation of pLDDT smaller than 15, which yields approximately 270K distilled samples. Moreover, we use representative protein structures for each cluster in AFESM (Yeo et al., 2025). We filter these structures with pLDDT larger than 0.8 resulting in more than 1.9M distilled strictures. All SimpleFold models except the largest 3B model are trained on the combination of three datasets listed above, adding up to approximately 2M structures.

To train our biggest model SimpleFold-3B, we explore an extended version AFESM (which we call AFESM-E) by also including structures beyond the cluster representatives. In particular, for each cluster, we randomly pick a maximum of 10 proteins structures with average pLDDT larger than 80, which resulting in a total of 8.6M distilled structures. Since larger models with larger capacity benefit from larger training sets, we train our largest SimpleFold-3B on the distilled AFESM-E data together with PDB and SwissProt.

## 3 RELATED WORK

**Protein Folding** Since the development of AlphaFold2 (Chakravarty & Porter, 2022) and RoseTTAFold (Baek et al., 2021) which achieved groundbreaking performance in protein folding with learning-based methods, many works have continued to investigate this problem (Ahdritz et al., 2024; Baek et al., 2023; Li et al., 2022). AlphaFold2 introduced domain specific modules like triangle attention, explicit modeling of pair representations, and MSA to extract evolutionary information of proteins. OmegaFold (Wu et al., 2022) and ESMFold (Lin et al., 2023b) replaced MSA with learned embeddings from pretrained PLM, which are efficient in inference and especially beneficial for orphan proteins. Some works also aimed at accelerating the models through efficient implementations of AlphaFold2 modules, like FastFold (Cheng et al., 2022) and MiniFold (Wohlwend et al.). These folding models are built on regression objectives and lack diversity for ensemble generation.

**Flow-Matching for Proteins** Generative models, especially diffusion and flow-matching based methods, have been introduced to protein modeling. AlphaFlow/ESMFlow (Jing et al., 2024a) proposed to tune AlphaFold2/ESMFold with flow-matching objectives and demonstrated advantages in ensemble generation. AlphaFold3 (Abramson et al., 2024) and its architectural reproductions (e.g., Boltz-1 (Wohlwend et al., 2024), Protenix (Team et al., 2025), Chai-1 (Boitreaud et al., 2024)) also used diffusion to build generative models for protein complexes of biomolecular interactions. In addition, several works have investigated diffusion or flow-matching models for de novo protein structure generation with heuristic architectural designs from AlphaFold, like RFDiffusion (Watson et al., 2023a), Genie-2 (Lin et al., 2024), P(all-atom) (Qu et al., 2024). (Jing et al., 2023) also developed crafted equivariant diffusion process. Proteina (Geffner et al., 2025) attempts to build a simplified architecture but still explicitly applies pair representation, and it only models $C_\alpha$ generation. In a strong departure from previous protein folding models, SimpleFold aims at tackling the folding problem with a general purpose transformer backbone and learning symmetries in the underlying data generation process directly from training data (Wang et al., 2023).

## 4 EXPERIMENTS

### 4.1 EXPERIMENTAL SETTINGS

We train a family of SimpleFold models at different sizes (i.e., 100M, 360M, 700M, 1.1B, 1.6B, and 3B) to investigate the scaling ability of proposed framework in folding (see detailed configurations in Tab. 5). The overall training of SimpleFold consistent of two training stages pre-training and finetuning. During the pre-training stage of SimpleFold we use a large dataset containing as much available data as possible. Finetuning, on the other hand, is performed on high-quality data to increase the fidelity of generated structures (see details in Appendix C.1).

### 4.2 PROTEIN FOLDING

We evaluate SimpleFold on two widely adopted folding benchmarks: CAMEO22 and CASP14, which are rigorous tests for generalization, robustness, and atomic-level accuracy in folding models. We set $\tau = 0.01$ for SimpleFold in inference which empirically shows best general performance in folding. We report standard structure prediction metrics: TM-score and GDT-TS assess global structural similarity; LDDT and LDDT-$C_\alpha$ measure local atomic accuracy across all atoms and $C_\alpha$ atoms, respectively; RMSD measures the averaged distance of atomic positions between two superimposed structures. For each metric, we report both the mean and the median score over all the test samples (separated by slashes). We report all the metrics for all-atom models and only report TM-score and GDT-TS for backbone-only models (see details in Appendix D).

Table 9 summarizes results on CASP14 and CAMEO22. We group approaches based on strategies to encode protein sequence, namely MSA or protein language model (PLM). For example, AlphaFold2 is MSA-based while ESMFold leverages PLM in place of MSA search. We also color baselines based on whether they are trained with generative objectives, i.e., diffusion / flow-matching or autoregression instead of direct regression to ground truth structures, e.g., AlphaFlow and ESMFlow are flow-matching models finetuned from AlphaFold2 and ESMFold, respectively.

Despite its simplicity, SimpleFold achieves competitive performance compared with these baselines. In both benchmarks, SimpleFold shows consistently better performance than ESMFlow which is also a flow-matching model built with ESM embeddings. On CAMEO22, SimpleFold demonstrates comparable results to the best folding models (e.g., ESMFold, RoseTTAFold2, and AlphaFold2). In particular, SimpleFold achieves over 95% performance of RoseTTAFold2/AlphaFold2 on most metrics without applying expensive and heuristic triangle attention and MSA. On the more challenging CASP14 benchmark, SimpleFold achieves even better performance than ESMFold. In particular, SimpleFold-3B obtains a TM-score of 0.720 / 0.792 and GDT-TS of 0.639 / 0.703 in comparison to 0.701 / 0.792 and 0.622 / 0.711 of ESMFold. SimpleFold also shows competitive or even better performance to baselines that applies MSA like RoseTTAFold and AlphaFlow.

Moreover, scaling up the model sizes of SimpleFold models results in better performance across the board, which indicates the benefit of designing a general purpose approach that benefits from scale. It is notable that scaling up model sizes improves performance substantially more in CASP14, i.e

Table 1: Performance of protein folding on the CAMEO22 (top) and CASP14 (bottom) benchmarks. For each metric, we report the average / median over all samples. Here, orange denotes baselines trained with regression objectives, green denotes baselines trained with generative objectives (i.e., diffusion/flow-matching or autoregression), and blue denotes our SimpleFold, which is trained with generative objective but without MSA.

| Type | Model | TM-score ↑ | GDT-TS ↑ | LDDT ↑ | LDDT-$C_\alpha$ ↑ | RMSD ↓ |
|---|---|---|---|---|---|---|
| | | *CAMEO22* | | | | |
| MSA-based | RoseTTAFold (Baek et al., 2021) | 0.780 / 0.860 | 0.715 / 0.775 | 0.575 / 0.605 | 0.798 / 0.827 | 5.721 / 2.864 |
| | AlphaFlow (Jing et al., 2024a) | 0.840 / 0.927 | 0.808 / 0.853 | 0.741 / 0.798 | 0.855 / 0.893 | 3.846 / 2.122 |
| | AlphaFold2 (Jumper et al., 2021) | 0.863 / 0.942 | 0.844 / 0.903 | 0.816 / 0.856 | 0.893 / 0.923 | 3.578 / 1.857 |
| | RoseTTAFold2 (Baek et al., 2023) | 0.864 / 0.947 | 0.845 / 0.904 | 0.727 / 0.767 | 0.893 / 0.926 | 3.571 / 1.707 |
| PLM-based | ESM3 (Hayes et al., 2025) | 0.746 / 0.840 | 0.694 / 0.758 | – | – | – |
| | ESMDiff (Lu et al., 2024a) | 0.754 / 0.847 | 0.701 / 0.760 | – | – | – |
| | EigenFold (Jing et al., 2023) | 0.750 / 0.840 | 0.710 / 0.790 | – | – | – |
| | OmegaFold (Wu et al., 2022) | 0.805 / 0.899 | 0.767 / 0.844 | 0.746 / 0.815 | 0.829 / 0.892 | 5.294 / 2.622 |
| | ESMFlow (Jing et al., 2024a) | 0.818 / 0.893 | 0.774 / 0.832 | 0.696 / 0.745 | 0.827 / 0.867 | 4.528 / 2.693 |
| | ESMFold (Lin et al., 2023b) | 0.853 / 0.933 | 0.826 / 0.875 | 0.792 / 0.834 | 0.871 / 0.906 | 3.973 / 2.019 |
| Ours | SimpleFold-100M | 0.803 / 0.878 | 0.746 / 0.787 | 0.721 / 0.752 | 0.822 / 0.852 | 4.897 / 2.855 |
| | SimpleFold-360M | 0.826 / 0.905 | 0.782 / 0.841 | 0.773 / 0.803 | 0.844 / 0.878 | 4.775 / 2.681 |
| | SimpleFold-700M | 0.829 / 0.915 | 0.788 / 0.845 | 0.775 / 0.809 | 0.850 / 0.886 | 4.557 / 2.423 |
| | SimpleFold-1.1B | 0.833 / 0.924 | 0.793 / 0.851 | 0.776 / 0.807 | 0.850 / 0.883 | 4.350 / 2.334 |
| | SimpleFold-1.6B | 0.835 / 0.916 | 0.799 / 0.864 | 0.782 / 0.816 | 0.853 / 0.889 | 4.397 / 2.187 |
| | SimpleFold-3B | 0.837 / 0.916 | 0.802 / 0.867 | 0.773 / 0.802 | 0.852 / 0.884 | 4.225 / 2.175 |
| | | *CASP14* | | | | |
| MSA-based | RoseTTAFold (Baek et al., 2021) | 0.654 / 0.678 | 0.562 / 0.572 | 0.464 / 0.456 | 0.705 / 0.723 | 9.676 / 6.420 |
| | AlphaFlow (Jing et al., 2024a) | 0.740 / 0.812 | 0.661 / 0.711 | 0.632 / 0.662 | 0.767 / 0.799 | 7.091 / 3.949 |
| | RoseTTAFold2 (Baek et al., 2023) | 0.802 / 0.881 | 0.740 / 0.824 | 0.638 / 0.669 | 0.824 / 0.869 | 6.744 / 3.292 |
| | AlphaFold2 (Jumper et al., 2021) | 0.845 / 0.907 | 0.783 / 0.855 | 0.778 / 0.817 | 0.856 / 0.897 | 5.027 / 3.015 |
| PLM-based | ESMDiff (Lu et al., 2024a) | 0.521 / 0.499 | 0.447 / 0.430 | – | – | – |
| | ESM3 (Hayes et al., 2025) | 0.534 / 0.567 | 0.459 / 0.488 | – | – | – |
| | EigenFold (Jing et al., 2023) | 0.590 / 0.637 | 0.539 / 0.575 | – | – | – |
| | ESMFlow (Jing et al., 2024a) | 0.627 / 0.679 | 0.539 / 0.544 | 0.525 / 0.539 | 0.669 / 0.730 | 10.503 / 6.974 |
| | OmegaFold (Wu et al., 2022) | 0.693 / 0.773 | 0.625 / 0.723 | 0.627 / 0.726 | 0.715 / 0.824 | 9.845 / 4.042 |
| | ESMFold (Lin et al., 2023b) | 0.701 / 0.792 | 0.622 / 0.711 | 0.637 / 0.705 | 0.725 / 0.802 | 8.679 / 4.016 |
| Ours | SimpleFold-100M | 0.611 / 0.628 | 0.513 / 0.544 | 0.537 / 0.549 | 0.659 / 0.685 | 11.157 / 8.976 |
| | SimpleFold-360M | 0.674 / 0.758 | 0.585 / 0.654 | 0.617 / 0.657 | 0.703 / 0.762 | 9.382 / 4.828 |
| | SimpleFold-700M | 0.680 / 0.767 | 0.591 / 0.668 | 0.630 / 0.674 | 0.714 / 0.763 | 9.289 / 4.431 |
| | SimpleFold-1.1B | 0.697 / 0.796 | 0.607 / 0.668 | 0.640 / 0.676 | 0.723 / 0.758 | 9.249 / 4.462 |
| | SimpleFold-1.6B | 0.712 / 0.801 | 0.630 / 0.709 | 0.660 / 0.699 | 0.741 / 0.798 | 8.424 / 4.722 |
| | SimpleFold-3B | 0.720 / 0.792 | 0.639 / 0.703 | 0.666 / 0.709 | 0.747 / 0.829 | 7.732 / 3.923 |

the more challenging benchmark, than in CAMEO22. This is a clear empirical evidence that models with larger capacity are more capable of solving complex folding tasks.

## 4.3 ENSEMBLE GENERATION

### 4.3.1 MOLECULAR DYNAMIC ENSEMBLE

SimpleFold trivially models the distribution of protein structures, due its generative training objective. Namely, SimpleFold does not only generate one deterministic structure for an input AA sequence but is also capable of generating the ensemble of different conformations. To demonstrate this ability of SimpleFold, we benchmark the performance on the ATLAS dataset (Vander Meersche et al., 2024), which assess generation of molecular dynamic (MD) ensemble structures. We set $\tau = 0.6$ (Eq. 4) in inference to add more stochasticity than folding tasks.

Compared to baselines without additional tuning on MD simulation data in ATLAS (e.g., MSA-subsampling), SimpleFold achieves superior performance on generating ensembles that match the distribution from MD simulations. We also report the results of SimpleFold-MD, a finetuned model on the training data split of ATLAS, comparing to baselines that are also additionally tuned (i.e., ESMDiff (Lu et al., 2024a), ESMFlow-MD (Jing et al., 2024a), and AlphaFlow-MD (Jing et al., 2024a)). As shown in Tab. 2, SimpleFold consistently achieves better performance than ESMFlow-MD where both rely on the ESM embedding without MSA. SimpleFold also shows better perfor-

Table 2: Evaluation on MD ensembles. Results of baseline models are taken from (Jing et al., 2024a; Lu et al., 2024a), to which the evaluation pipeline for our SimpleFold (SF) and SimpleFold-MD (SF-MD) adheres.

| | No Tuning | | | Tuned | | | |
| --- | --- | --- | --- | --- | --- | --- | --- |
| | AF2 | MSA-sub. | SimpleFold | ESMDiff | ESMFlow-MD | AlphaFlow-MD | SimpleFold-MD |
| **Pairwise RMSD r** ↑ | 0.10 | 0.22 | **0.44** | 0.18 | 0.19 | **0.48** | 0.45 |
| **Global RMSF r** ↑ | 0.21 | 0.29 | **0.45** | 0.49 | 0.31 | **0.60** | 0.48 |
| **Per target RMSF r** ↑ | 0.52 | 0.51 | **0.60** | 0.68 | 0.76 | **0.85** | 0.67 |
| **RMWD** ↓ | **3.58** | 4.28 | 4.22 | 7.48 | 3.60 | **2.61** | 4.17 |
| **RMWD trans contri** ↓ | **2.86** | 3.33 | 3.74 | 5.18 | 3.13 | **2.28** | 3.40 |
| **RMWD var contri** ↓ | 2.27 | 2.24 | **1.74** | 3.37 | 1.74 | **1.30** | 1.88 |
| **MD PCA W2** ↓ | 1.99 | 2.23 | **1.62** | 2.29 | 1.51 | 1.52 | **1.34** |
| **Joint PCA W2** ↓ | 2.86 | 3.57 | **2.59** | 6.32 | 3.19 | **2.18** | 2.85 |
| **% PC sim > 0.5** ↑ | 23 | 21 | **37** | 23 | 26 | **44** | 38 |
| **Weak contacts J** ↑ | 0.27 | 0.37 | 0.36 | 0.52 | 0.55 | **0.62** | 0.56 |
| **Transient contacts J** ↑ | **0.28** | 0.27 | **0.27** | 0.26 | 0.34 | **0.41** | 0.34 |
| **Exposed residue J** ↑ | 0.32 | 0.37 | **0.39** | - | 0.49 | 0.50 | **0.60** |
| **Exposed MI matrix** $\rho$ ↑ | 0.02 | 0.10 | **0.14** | - | 0.20 | 0.25 | **0.32** |

Table 3: Two-state conformation results. For the last two metrics, both mean and median are reported over the targets. Results are taken from the ESMDiff paper (Lu et al., 2024a), to which the evaluation pipeline for the rest models adhere.

| Type | Model | Res. flex. (global) ↑ | Res. flex. (per-target) ↑ | TM-ens ↑ | Res. flex. (global) ↑ | Res. flex. (per-target) ↑ | TM-ens ↑ |
| --- | --- | --- | --- | --- | --- | --- | --- |
| | | | Apo/holo | | | Fold-switch | |
| Seq-based | FoldFlow2 (Huguet et al., 2024) | 0.027 | 0.057 / 0.055 | 0.216 / 0.208 | 0.051 | 0.009 / 0.005 | 0.199 / 0.191 |
| | MultiFlow (Campbell et al., 2024) | 0.113 | 0.211 / 0.194 | 0.360 / 0.342 | 0.092 | 0.068 / 0.061 | 0.269 / 0.250 |
| | Str2Str (Lu et al., 2024b) | 0.174 | 0.326 / 0.307 | 0.731 / 0.728 | 0.161 | 0.246 / 0.233 | 0.615 / 0.644 |
| | Eigenfold (Jing et al., 2023) | 0.126 | 0.407 / 0.401 | 0.830 / 0.870 | 0.225 | 0.279 / 0.255 | 0.614 / 0.653 |
| | ESMDiff (Lu et al., 2024a) | 0.420 | 0.489 / 0.515 | 0.838 / 0.877 | **0.402** | 0.341 / 0.288 | 0.626 / 0.685 |
| | ESMFlow (Jing et al., 2024a) | 0.416 | 0.496 / 0.522 | 0.856 / 0.893 | 0.269 | 0.345 / 0.329 | 0.700 / 0.755 |
| MSA-based | MSA-Subs. (Jumper et al., 2021) | 0.398 | 0.404 / 0.371 | 0.856 / 0.894 | 0.350 | 0.320 / 0.303 | 0.714 / 0.765 |
| | AlphaFlow (Jing et al., 2024a) | 0.455 | 0.527 / 0.527 | 0.864 / 0.893 | 0.385 | **0.384 / 0.376** | **0.730 / 0.788** |
| Ours | SimpleFold-100M | 0.492 | 0.500 / 0.532 | 0.852 / 0.887 | 0.391 | 0.291 / 0.241 | 0.656 / 0.677 |
| | SimpleFold-360M | 0.537 | 0.520 / 0.528 | 0.864 / 0.898 | 0.359 | 0.310 / 0.314 | 0.689 / 0.746 |
| | SimpleFold-700M | 0.552 | 0.524 / 0.538 | 0.870 / 0.899 | 0.307 | 0.328 / 0.310 | 0.693 / 0.713 |
| | SimpleFold-1.1B | 0.557 | 0.526 / 0.537 | 0.870 / 0.900 | 0.337 | 0.346 / 0.344 | 0.698 / 0.755 |
| | SimpleFold-1.6B | 0.501 | 0.522 / 0.508 | 0.877 / 0.912 | 0.240 | 0.339 / 0.318 | 0.721 / 0.770 |
| | SimpleFold-3B | **0.639** | **0.550 / 0.552** | **0.893 / 0.916** | 0.292 | 0.288 / 0.263 | **0.734 / 0.766** |

mance than AlphaFlow-MD on metrics related to ensemble observables (e.g., exposed residue and MI matrix), which are a key feature in the identification of cryptic pockets in drug discovery.

### 4.3.2 MULTI-STATE STRUCTURE PREDICTION

We also evaluate the capacity of SimpleFold in generating protein structures with multiple natural conformation. We adopt the benchmarking set of *Apo/holo* (Saldaño et al., 2022) and *Fold-switch* (Chakravarty & Porter, 2022) following Jing et al. (2023). The model is assessed to produce a diverse yet accurate set of samples "covering" both conformational states and reflecting correct local flexibility. As shown in Tab. 3, SimpleFold obtains state-of-the-art performance on *Apo/holo*, where SimpleFold outperforms strong MSA-based approaches like AlphaFlow significantly. On Fold-switch, SimpleFold shows comparable or even better performance than ESMFlow which is also applies flow-matching objective and is built on ESM embeddings. The results validate the capability of our SimpleFold in predicting the structures of high quality (i.e., ensemble TM-score) as well as correctly modeling the flexibility in structures (i.e., residue flexibility). Also, the overall performance of SimpleFold increases with the model size growing, which further showcase potential of our proposed framework in generating protein ensembles.

### 4.4 EFFECTS OF SCALING IN PROTEIN FOLDING

SimpleFold benefits from increasing model sizes as proven by recent success of generative models in vision and language generation. We note that the effects of scaling both training data and model

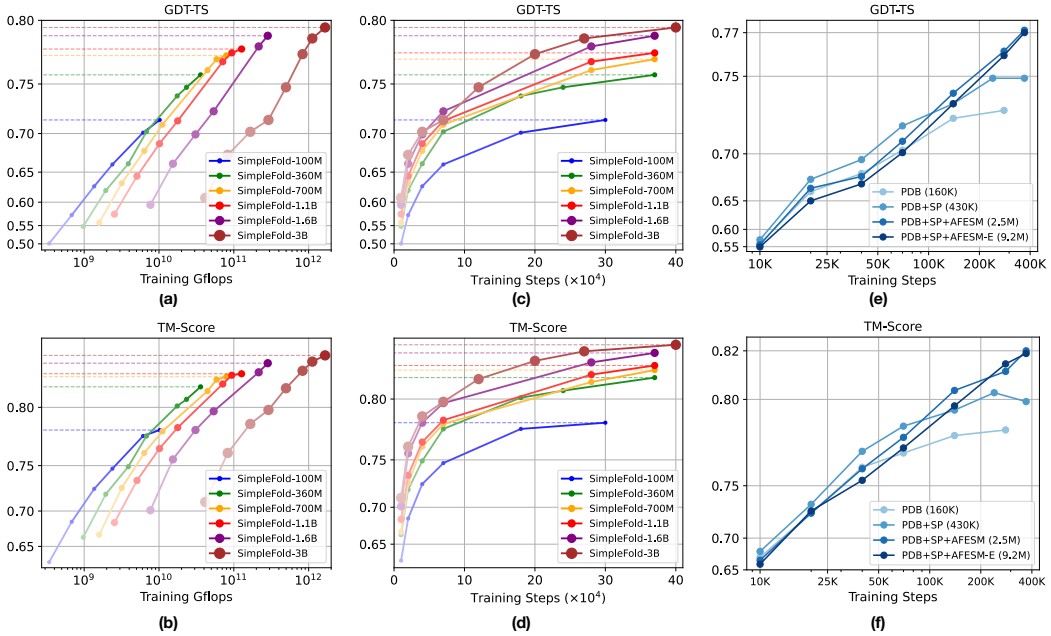

Figure 3: Scaling behavior of SimpleFold. Training Gflops vs. folding performance on GDT-TS and (b) TM-score. Training steps vs. folding performance on (c) GDT-TS and (d) TM-score. How data scale affects the performance (e) GDT-TS and (f) TM-score. All models are benchmarked on CAMEO22.

sizes have not yet been rigorously investigated in protein folding. The section empirically shows the scaling behavior of SimpleFold from both model and data perspectives, highlighting important considerations for building powerful biological generative models. We train models with different sizes from the smallest with 100M parameters to the largest with 3B parameters on full pre-training data containing PDB, SwissProt from AFDB, and AFESM (AFESM-E for 3B model). Fig. 3(a)-(d) illustrate how model sizes affect folding performance (also see Fig. 1(d)). Larger models trained with a larger training budget (i.e., training Gflops and training iterations) achieve better performance. We believe these results highlight the positive scaling behavior of SimpleFold and an direction of progress to obtain more powerful generative models in biology.

We also show the benefits of scaling up training data in SimpleFold. We train SimpleFold-700M with different sources of training data. As shown in Fig. 3(e) and (f), SimpleFold when increasing the total number of unique structures in the data mix, the final performance of SimpleFold tends to improve after sufficient training iterations. These experimental results support our core contribution to build a simplified and scalable folding model that benefits from the growing total of protein data available either experimentally or distilled from different models.

## 5 CONCLUSIONS AND FUTURE WORK

We introduced SimpleFold, a flow-matching generative model for protein folding that represents a strong departure from the architectural designs in previous approaches. SimpleFold is solely built with general-purpose transformer blocks with adaptive layers, dispensing away with heuristic designs like expensive pair representations and triangular updates introduced by AlphaFold2. This simplified framework allows us to train SimpleFold at scale both in terms of model size and training data. Our largest (and most powerful) model, SimpleFold-3B, demonstrates competitive performance on standard folding tasks and it also show very strong or even state-of-the-art results on ensemble generation tasks. To the best of our knowledge, SimpleFold is the first work that rigorously demonstrates good scaling behavior in protein folding. We believe SimpleFold represents a

disruptive approach for protein folding that relies on scaling up general purpose architecture blocks to learn the symmetries of the underlying data generation process directly from training data.

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

## A    DATA PIPELINE

We largely adopt the data pipeline implemented in Boltz-1[1] (Wohlwend et al., 2024), which is an open-source replication of AlphaFold3 (Abramson et al., 2024). Tab. 4 lists the input features for SimpleFold. It is noted that since SimpleFold does not apply MSA or template search, input features are also simplified compared to AlphaFold.

In cropping larger proteins, we follow a cropping algorithm that combines both spatial and contiguous cropping strategies introduced in previous work (Chakravarty & Porter, 2022; Abramson et al., 2024; Wohlwend et al., 2024). Following this setting, we set the neighborhood size in cropping uniformly between zero and 40 tokens to balance spatial and contiguous cropping.

Table 4: Input features to SimpleFold.

| Feature | Shape | Description |
|---|---|---|
| residue_index | $[N_r]$ | Residue number in the token's original input chain. |
| token_index | $[N_r]$ | Token number. Increases monotonically. |
| restype | $[N_r]$ | One-hot encoding of the sequence: 20 amino acids + unknown. |
| esm_embed | $[N_r, 37, 2560]$ | Protein sequence embedding from all layers in ESM2-3B. |
| noised_pos | $[N_a, 3]$ | Noised atom positions, $\mathbf{x}_t$ in Å (random rotation applied). |
| ref_pos | $[N_a, 3]$ | Atom positions in the reference conformer in Å (no rotation applied). |
| ref_mask | $[N_a]$ | Mask indicating atoms used in the reference conformer. |
| ref_element | $[N_a, 128]$ | One-hot encoding of the element number for each atom. |
| ref_charge | $[N_a]$ | Charge for each atom in the reference conformer. |
| ref_atom_name_chars | $[N_a, 4, 64]$ | One-hot encoding of atom names in the reference conformer. |
| ref_space_uid | $[N_a]$ | Encoding of the residue index associated with reference conformer. |
| time | $[1]$ | Timestep in flow process. |
| length | $[1]$ | Number of residues, $N_r$. |

During training, atomic positions of a protein are mean centered and augmented with random rotation. After centering, we scale the position by global factor of $1/16$ to make the atomic positions live in the $[-1, 1]$ interval. Similarly, we also scale ref_pos by $1/5$ to standardize the positions in reference conformers.

## B    MODEL ARCHITECTURE

### B.1    ARCHITECTURE COMPARISON TO ALPHAFOLD2

Fig. 4 depicts the comparison of major compute blocks in AlphaFold2 and SimpleFold (Fig. 4(a) borrowed from original AlphaFold2 paper (Chakravarty & Porter, 2022)). As shown in the figure, SimpleFold does not rely on either explicit pair representations or MSA. Instead, we only keep a sequence-level representation and leverage embeddings extracted from pretrained PLM (i.e., ESM2 (Lin et al., 2023b)). Compared AlphaFold's Evoformer block which includes expensive triangle attention to interact between pair and sequence representations, SimpleFold follows a simple DiT architecture (Peebles & Xie, 2023) which is more computationally efficient.

### B.2    COMPARISON TO ALPHAFLOW AND ESMFLOW

Though SimpleFold, AlphaFlow and ESMFlow (Jing et al., 2024a) all use a flow-matching training objectives, the architectural design and the training paradigm are drastically different: the architectural design and the training paradigm are drastically different.

AlphaFlow and ESMFlow are built upon the AlphaFold (Chakravarty & Porter, 2022) and ESM-Fold (Lin et al., 2023b) network architectures, respectively. This means they inherit domain-specific heuristic architectural designs like pair representation and triangle attention. On the other hand, SimpleFold is based purely on standard transformer blocks without any domain-specific network

---

[1]`https://github.com/jwohlwend/boltz`

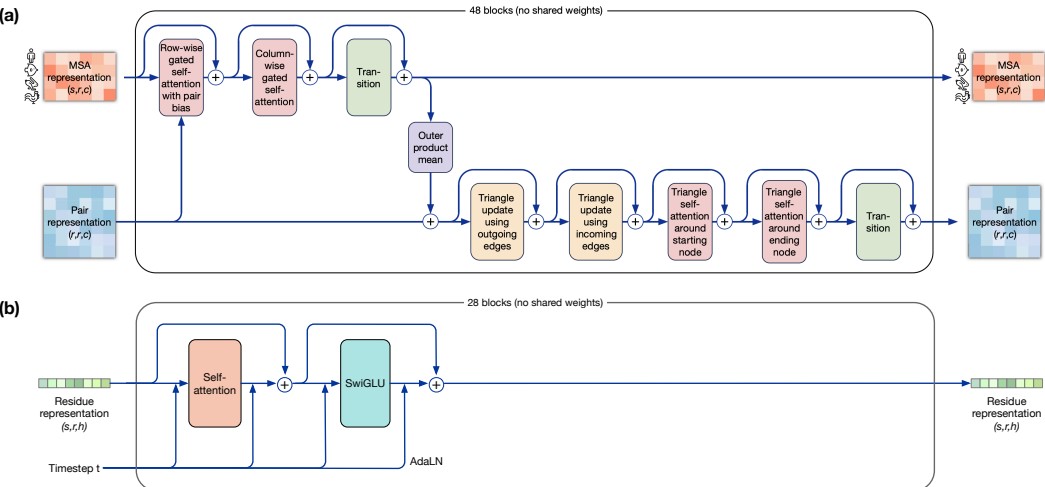

Figure 4: Major neural network blocks of (a) Evoformer in AlphaFold2, and (b) Transformer with adaptive layer in SimpleFold.

blocks. In addition, AlphaFlow and ESMFlow use a generative training objective merely as a fine-tuning strategy on top of already fully trained checkpoints from AlphaFold2 and ESMFold which use a deterministic regression objective. On the contrary, SimpleFold is built from the ground up to be a pure generative model trained from scratch with a flow-matching objective.

Building SimpleFold from the ground up as a generative model that is trained from scratch with a flow-matching objective results in improvements in multi-state benchmarks over models that only fine-tune pre-trained deterministic models like AlphaFold and ESMFold.

### B.3 IMPLEMENTATION DETAILS

We adopt a modern implementation stack for all the transformer blocks including QK-normalization (Esser et al., 2024) and SwiGLU (Shazeer, 2020) in place of standard FFN for better performance and training stability. To encode the positional information of atoms and residues, we employ rotary position embedding (RoPE) (Su et al., 2024). Particularly in each attention block within the residue trunk, they query and key vectors of the $n$-th residue in a amino acid sequence are rotated by $e^{i\theta n}$. In both the atom encoder and decoder, we extend the positional embedding to a 4D axial RoPE. The first three axes are 3D atomic coordinates from reference conformers (see Appendix A), which are local structures predicted at the amino acid level by a rule-based cheminformatic method. The last axis is the 1D indexing to the corresponding residue token. Each axis in 4D axial RoPE controls rotation of a quarter of the hidden dimension in both query and key.

### B.4 MODEL CONFIGURATIONS

Table 5 lists the configurations of different SimpleFold models from the smallest 94M to largest 2.86B. In implementation, we apply the same architecture for the atom encoder and atom decoder. Though AlphaFold2 is similar to our smallest SimpleFold-100M in terms of number of parameters (both are around 95M), its forward Gflops are much higher than our largest SimpleFold-3B ($\sim$ 30Tflops vs. $\sim$ 1.4Tflops). This is because AlphaFold2 relies on expensive triangle update as well as explicit modeling pair representations from MSA. SimpleFold, on the other hand, is built on general-purposed transformer blocks which are much more computationally efficient.

Table 5: Configurations of different variants of SimpleFold with comparison to AlphaFold2 and ESMFold in number of parameters and forward Gflops.

| Model | # Params | Gflops | Atom Enc. / Dec. Dim. | # Heads | # Blocks | Residue Trunk Dim. | # Heads | # Blocks |
|---|---|---|---|---|---|---|---|---|
| AlphaFold2 | 95M | 30935.0 | - | - | - | - | - | - |
| ESMFold | 710M | 3399.7 | - | - | - | - | - | - |
| SimpleFold-100M | 94M | 66.5 | 256 | 4 | 1 | 768 | 12 | 8 |
| SimpleFold-360M | 360M | 189.9 | 256 | 4 | 2 | 1024 | 16 | 18 |
| SimpleFold-700M | 687M | 310.4 | 256 | 4 | 2 | 1152 | 16 | 28 |
| SimpleFold-1.1B | 1.11B | 496.0 | 384 | 6 | 2 | 1280 | 20 | 36 |
| SimpleFold-1.6B | 1.58B | 750.0 | 512 | 8 | 3 | 1536 | 24 | 36 |
| SimpleFold-3B | 2.86B | 1382.4 | 640 | 10 | 4 | 2048 | 32 | 36 |

## B.5 GROUPING AND UNGROUPING

Fig. 5 illustrates how grouping and ungrouping operations are conducted in SimpleFold. In grouping, we conduct average pooling over atoms tokens from one residue to obtain a residue token. While in ungrouping, we replicate the same updated residue tokens to all atoms within the residue.

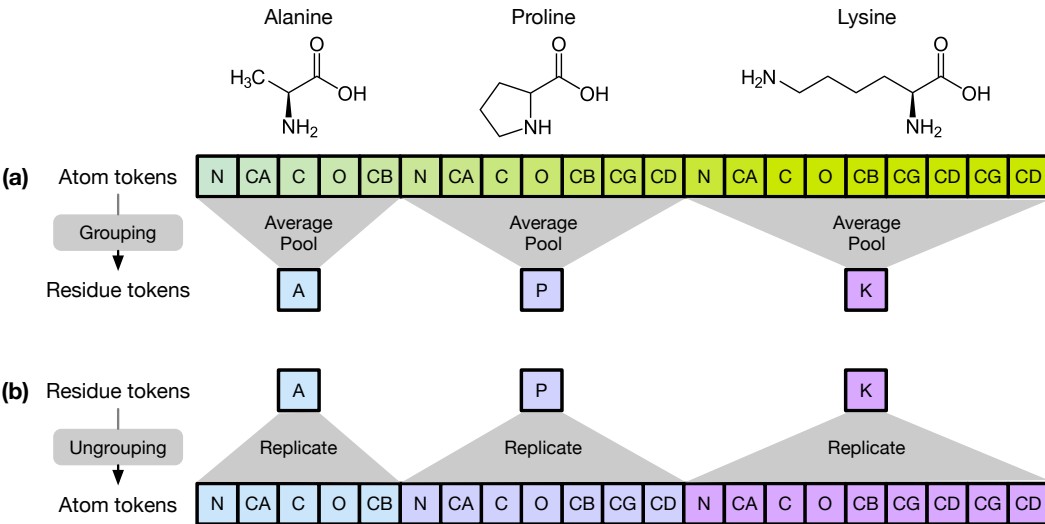

Figure 5: Illustration of (a) grouping and (b) ungrouping operations in SimpleFold.

## C TRAINING AND INFERENCE

### C.1 ADDITIONAL TRAINING DETAILS

**Pre-training.** In pre-training, SimpleFold is trained on structures from all three data sources, namely PDB, SwissProt from AFDB, and AFESM (AFESM-E for 3B model). We set the maximal amino acid sequence length to 256, where we keep shorter sequence without padding while crop longer sequences to 256 residues. We set $\alpha(t) = 1$ in Eq. 3. All models are trained with effective batch size 512 except for 1.6B and 3B models which are trained with batch size 1024 and 3072, respectively. We use the AdamW optimizer (Loshchilov & Hutter, 2019) with learning rate 0.0001 and linear warmup for the first 5000 steps. We also apply SO(3) data augmentation during training, which randomly rotates structure targets, and rely on the capacity of the model to directly learn such symmetries during training.

**Finetuning.** In finetuning, SimpleFold is trained on PDB and SwissProt subsets only which contain higher quality data. We set a maximal sequence length to 512 which allows access to

larger protein structures in this training phase and accordingly half effective batch size. We set $\alpha(t) = 1 + 8\texttt{ReLU}(t - 0.5)$ in Eq. 3 which gradually increases weight of LDDT loss to maximum value of 5 when approaching clean data ($t = 1$). We keep AdamW as an optimizer with the same learning rate $0.0001$ in finetuning. In both pre-training and finetuning, we apply an exponential moving average (EMA) of all model weights with a decay of $0.999$ following a common practice in flow-matching generative models.

**Timestep Resampling.** In training, we resample timestep with $p(t) = 0.02\,\mathcal{U}(0, 1) + 0.98\,\texttt{LN}(0.8, 1.7)$, and logit-normal distribution $\texttt{LN}$ is given:

$$\texttt{LN}(t; m, s) = \frac{1}{t(1 - t)s\sqrt{2\pi}} \exp -\frac{(\texttt{logit}(t)) - m)^2}{2s^2}. \tag{5}$$

We set $m = 0.8, s = 1.7$ to sample timestep more densely around $t = 1$ so the model better learns to capture the refined details as shown in Fig 6.

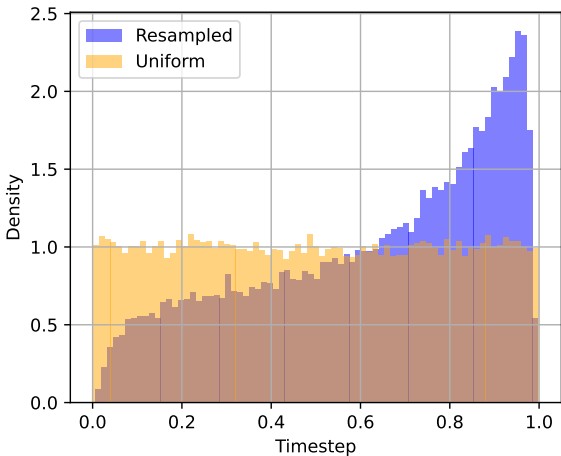

Figure 6: Distribution of resampled timestep compared to uniform distribution.

**Rigid alignment.** Following Abramson et al. (2024); Wohlwend et al. (2024), we apply a rigid alignment between one-step denoising atomic coordinates and true coordinates before computing the flow-matching MSE loss (Eq. 1) in training to reduce the loss variance. In particular, $\hat{\mathbf{x}}(\mathbf{x}_t)$ is estimated through one step Euler, i.e., $\hat{\mathbf{x}}(\mathbf{x}_t) = \mathbf{x}_t + (1 - t)\,\mathbf{v}_\theta(\mathbf{x}_t, \mathbf{s}, t)$, and the true coordinates $\mathbf{x}$ is aligned with the denoised coordinates through Kabsch algorithm (Wohlwend et al., 2024) to obtain $\mathbf{x}'$. The velocity target is re-calculated by interpolating the aligned $\mathbf{x}'$ and noise $\boldsymbol{\epsilon}$. Though such a rigid alignment strategy helps in faster convergence, it does not make a significant difference in final performance as also mentioned in Wohlwend et al. (2024).

**LDDT Loss.** Following AlphaFold3 (Abramson et al., 2024), the nonlinear function $\sigma$ in Eq. 2 is given as:

$$\sigma(x) = \frac{1}{4}(\texttt{sigmoid}(0.5 - x) + \texttt{sigmoid}(1 - x) + \texttt{sigmoid}(2 - x) + \texttt{sigmoid}(4 - x)), \tag{6}$$

which mimics the how LDDT is computed for evaluation. We set the cutoff distance $\mathcal{C} = 15\text{Å}$ in Eq. 2, which is the typical setting for the LDDT metric.

**Batching.** During training we copy one protein $B_c$ times per GPU with different flow timestep $t$ sampled and accumulate gradients from $B_p$ different proteins on different GPUs, following AlphaFold2 (Chakravarty & Porter, 2022; Abramson et al., 2024). Therefore, the effective batch size is $B_c \times B_p$. We empirically find that this strategy leads to a more stable gradient and better performance than naively building a batch with randomly selected proteins. Tab. 6 lists the detailed setting of training batch for different model sizes.

Table 6: Settings of pre-training and finetuning batches for different SimpleFold models.

| Model | Pre-training | | | Finetuning | | |
|---|---|---|---|---|---|---|
| | # Copies $B_c$ | # Prot. $B_p$ | Eff. Bsz. | # Copies $B_c$ | # Prot. $B_p$ | Eff. Bsz. |
| SimpleFold-100M | 16 | 32 | 512 | 8 | 32 | 256 |
| SimpleFold-360M | 16 | 32 | 512 | 8 | 32 | 256 |
| SimpleFold-700M | 16 | 32 | 512 | 8 | 32 | 256 |
| SimpleFold-1.1B | 16 | 32 | 512 | 8 | 32 | 256 |
| SimpleFold-1.6B | 8 | 128 | 1024 | 4 | 128 | 512 |
| SimpleFold-3B | 24 | 128 | 3072 | 12 | 128 | 1536 |

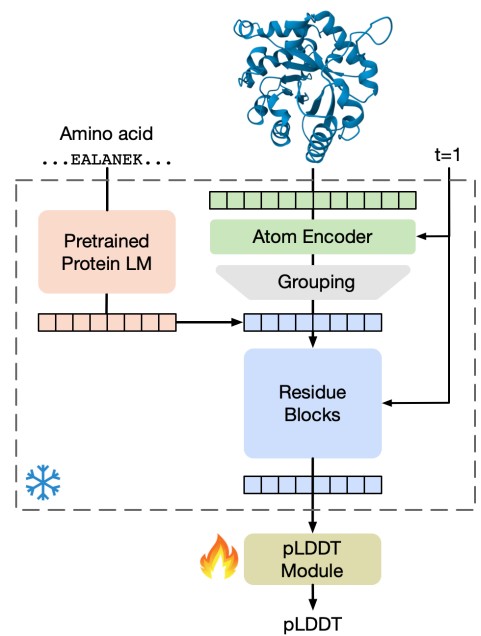

Figure 7: Illustration of pLDDT module training.

**Confidence Module.** Providing a confidence estimation for generated protein structures can greatly help understand the quality of generation (Chakravarty & Porter, 2022; Lin et al., 2023b). To this end, we develop an additional predicted LDDT (pLDDT) module which predicts a per-residue LDDT value (ranging from 0 to 100) as a confidence score. After the folding model is fully trained, we train the pLDDT module in a separate training stage while freezing all the parameters in the folding model (see Fig. 7). During training the pLDDT module, we sample protein structures $\hat{x}$ on the fly, and feed $\hat{x}$ into the folding model with timestep $t = 1$ for adaptive layers to acquire the final residue tokens $r$. The pLDDT module is composed of 4 layers of standard transformer blocks without adaptive layers, which takes in $r$ and outputs pLDDT. Following (Chakravarty & Porter, 2022), the target LDDT is discretized into 50 bins and the pLDDT module is trained through a cross-entropy objective.

After SimpleFold is pretrained and finetuned, we train the pLDDT module with all other components frozen. The pLDDT module is trained on combination of PDB and SwissProt data, which contains experimental and high-quality distilled data. During pLDDT training, we set $\alpha(t) = 1$, and Simple-Fold generates structure samples on the fly with 200 steps and $\tau = 0.3$. As in finetuning, We set maximal sequence length to 512 and apply AdamW optimizer with the learning rate 0.0001. Fig. 7 shows the training pipeline for pLDDT module. In particular, we use fully trained SimpleFold-1.6B to extract residue tokens.

## C.2 ADDITIONAL INFERENCE DETAILS

During inference, we use the Euler–Maruyama integrator shown in Eq. 4 starting from $t_\epsilon = 0.0001$ and the number of time steps is set to be 500 without additional statement. In practice, we set $\eta = 0.01$ in $w(t) = \frac{2(1-t)}{t+\eta}$ for numerical stability. And following (Geffner et al., 2025), we set $w(t) = 0$ for $t \geq 0.99$ and discretize the time interval logarithmically from $t = t_\epsilon$ to $t = 1$. After each sampler step, we rescale the center of all atomic positions to origin to align with the training setting. At the end of the flow trajectory, we rescale the coordinates by multiplying 16 to map the protein structure back to Å scale.

## C.3 INFERENCE TIME

Tab. 7 lists inference time of SimpleFold in comparison to baseline models, AlphaFold2, ESMFold, AlphaFlow, and ESMFlow. SimpleFold shows advantage in inference efficiency especially when sequence is longer (e.g., 1024). Also, ESM2 adds little overhead in inference.

Table 7: Inference time (in seconds) of different models. All models are benchmarked on a single H100 with batch size 1.

| | Steps/Recycles | Sequence length | | | | |
| --- | --- | --- | --- | --- | --- | --- |
| | | 64 | 128 | 256 | 512 | 1024 |
| AlphaFold2 | 3 | 3.0 | 3.7 | 8.0 | 25.5 | 111.5 |
| ESMFold | 3 | 1.100 | 1.1 | 1.745 | 7.4 | 43.6 |
| AlphaFlow | 10 | 10.0 | 12.3 | 26.6 | 85.2 | 371.7 |
| ESMFlow | 10 | 3.7 | 3.7 | 5.8 | 24.6 | 145.5 |
| ESM2 | 1 | <0.1 | <0.1 | 0.1 | 0.2 | 0.4 |
| SimpleFold-100M | 200 | 3.6 | 3.6 | 3.8 | 4.2 | 5.6 |
| SimpleFold-360M | 200 | 7.2 | 7.4 | 7.6 | 8 | 11.6 |
| SimpleFold-700M | 200 | 9.8 | 10.2 | 10.4 | 11.4 | 16.6 |
| SimpleFold-1.1B | 200 | 12.6 | 12.8 | 12.8 | 15 | 22.2 |
| SimpleFold-1.6B | 200 | 13.0 | 13.0 | 13.8 | 18.2 | 29.4 |
| SimpleFold-3B | 200 | 14.0 | 14.0 | 15.6 | 27.8 | 44.6 |
| SimpleFold-100M | 500 | 9.0 | 9.0 | 9.5 | 10.5 | 14 |
| SimpleFold-360M | 500 | 18.0 | 18.5 | 19 | 20 | 29 |
| SimpleFold-700M | 500 | 24.5 | 25.5 | 26 | 28.5 | 41.5 |
| SimpleFold-1.1B | 500 | 31.5 | 32.0 | 32 | 37.5 | 55.5 |
| SimpleFold-1.6B | 500 | 32.5 | 32.5 | 34.5 | 45.5 | 73.5 |
| SimpleFold-3B | 500 | 35.4 | 35.4 | 37.2 | 72.2 | 111.4 |

# D EVALUATION

## D.1 FOLDING BASELINES

**AlphaFold2.** The AlphaFold2 (AF2) baseline was established using the official implementation wrapped using ColabFold (Mirdita et al., 2022). We utilized the standard released weights with three model recycles. We adopt the MMSeqs2 engine (Steinegger & Söding, 2017) to search for multiple sequence alignments (MSAs) as model input. No template or `Amber` relax is applied to the predictions.

**RoseTTAFold.** We utilized the release models of RoseTTAFold (Baek et al., 2021) via Colab-Fold (Mirdita et al., 2022), employing its publicly available pre-trained model weights. We keep the default configurations of both models for inference and use MMseqs2(Steinegger & Söding, 2017) for MSA search. In specific, we use the proposed pipeline in the ColabFold Notebook including the end-to-end 3-track model forward, TRFold refinement and side-chain packing using SCRWL4 (Krivov et al., 2009).

**RoseTTAFold2.** Experiments of RoseTTAFold2 (Baek et al., 2023) are similarly conducted via ColabFold (Mirdita et al., 2022) with the pre-trained model weights. We follow the default inference configuration as described in the RoseTTAFold2 (Baek et al., 2023) official repository by setting `-n_recycles=3, -nseqs=256 and -subcrop=-1`.

**ESMFold.** Our experiments employed the ESMFold implementation from ColabFold (Mirdita et al., 2022) and model checkpoints from (Lin et al., 2023b). We used the pretrained `esmfold_v1` model for inference as recommended by the authors, the performance of which is better than the `esmfold_v0` model which was used for experiments in ESM2 paper (Lin et al., 2023b). We set the number of recycles to be 3, aligned with the AF2 setting.

**OmegaFold.** The implementation of OmegaFold used was based on the original repository (Wu et al., 2022). We relied on the default pre-trained model shipped with the release. The inference pipeline is strictly adhering to the default setting.

**EigenFold.** The EigenFold implementation as provided (Jing et al., 2023) was leveraged for our baseline runs. We utilized the standard pre-trained weights for decoder and make node/edge embeddings from OmegaFold as instructed by the authors. Defaults settings applied during inference included `-alpha 1 -beta 3 -elbo_step 0.2`.

**AlphaFlow/ESMFlow.** We utilized the codebase for AlphaFlow and ESMFlow released by the authors of (Jing et al., 2024a), employing its pre-trained model checkpoints on PDB data (with suffix `pdb_base_202402.pt`). The setup largely mirrored the default configurations for both models described in the repository, specifically by setting `tmax` to be 1.0 and the flow `steps` to be 10.

**ESM3/ESMDiff.** The implementation of ESMDiff and ESM3 used was based on the ESMDiff original repository (Lu et al., 2024a). No additional training was performed; the provided pre-trained model was used directly (both pretrained ESM3 and finetuned ESMDiff) to predict the structures for each target. We used standard hyperparameters as listed by the authors, including `num_steps=25, T=1.4, top_p=0.9`.

## D.2 PDB CUTOFF DATE

Tab. 8 lists the PDB cutoff date of most baselines in training. SimpleFold uses May 1, 2020 as the cutoff date following most baselines.

Table 8: Cutoff date of PDB for training.

| Model | PDB cutoff date |
|---|---|
| AlphaFold2 Chakravarty & Porter (2022) | May 1, 2018 |
| RoseTTAFold2 (Baek et al., 2023) | April 30, 2020 |
| ESMFold (Lin et al., 2023b) | May 1, 2020 |
| EigenFold (Jing et al., 2023) | April 30, 2020 |
| AlphaFlow (Jing et al., 2024a) | May 1, 2018 |
| ESMFlow (Jing et al., 2024a) | May 1, 2020 |
| ESM3 (Hayes et al., 2025) | May 1, 2020 |
| ESMDiff (Lu et al., 2024a) | May 1, 2020 |
| SimpleFold | May 1, 2020 |

## D.3 ESTIMATION OF GFLOPS

We leverage DeepSpeed (Rasley et al., 2020) library to estimate forward flops for SimpleFold as well as baseline models. In particular, we use `deepspeed.profiling.flops_profiler.get_model_profile` [2] function to get the compute profile for the models. In estimating the flops, we set the number of residues to be 256 and number of atoms to be 2304, namely, 9 atoms per residue.

## D.4 TARGETS IN FOLDING TASKS

List of 183 targets in CAMEO22 (Haas et al., 2018):

---

[2]`https://www.deepspeed.ai/tutorials/flops-profiler/`

```
7dz2-C, 7eoz-A, 7fac-A, 7fgb-A, 7fgp-A, 7fh0-B, 7lt7-A, 7lx4-A, 7m1z-B,
7mj3-A, 7n3y-A, 7n6h-A, 7n99-A, 7oj1-A, 7oj2-A, 7oju-A, 7pc1-A, 7pce-A,
7pcv-A, 7pk5-A, 7pkw-A, 7pl4-A, 7pl7-A, 7pqi-A, 7pqw-A, 7pup-A, 7pwe-A,
7q6d-A, 7q6g-A, 7q83-D, 7q9e-C, 7qau-A, 7qpe-A, 7qsw-B, 7qsw-C, 7qsx-A,
7qys-L, 7r08-E, 7r0o-B, 7r3w-D, 7r49-B, 7rlk-D, 7rmy-A, 7roa-A, 7rpn-A,
7rt7-D, 7rup-A, 7ruq-A, 7s03-A, 7s8k-B, 7sao-A, 7sbd-H, 7sfn-B, 7skh-B,
7skj-A, 7snc-A, 7snj-A, 7soo-A, 7spn-A, 7sz2-B, 7t12-B, 7t1j-B, 7t5w-B,
7te2-A, 7tgi-B, 7th2-C, 7tif-A, 7tol-A, 7tvw-A, 7u04-H, 7u0e-H, 7uav-A,
7ug9-A, 7upm-A, 7upv-A, 7uqv-D, 7uwg-C, 7ux0-A, 7uxt-A, 7v2s-B, 7v5f-A,
7v8t-A, 7vbo-A, 7vd7-B, 7vf3-B, 7vfc-A, 7vfq-C, 7vi8-B, 7vil-A, 7vma-A,
7vmf-A, 7vmh-C, 7vp3-C, 7vp6-D, 7vpu-A, 7vqk-A, 7vr2-A, 7vrf-A, 7vt4-A,
7vt5-A, 7vyu-A, 7w06-A, 7w16-A, 7w42-B, 7w52-B, 7w6x-A, 7w7h-E, 7w89-A,
7w8u-A, 7wa9-A, 7wbn-A, 7wf6-A, 7wf8-B, 7wf9-A, 7wfx-A, 7whf-G, 7wj0-A,
7wjt-B, 7wq5-A, 7wua-A, 7x0g-A, 7x0q-A, 7x0r-B, 7x15-A, 7x1k-A, 7x7w-A,
7x8c-B, 7xce-A, 7xjt-B, 7xtm-B, 7y0i-A, 7y39-B, 7y3k-A, 7y3w-A, 7y4n-A,
7y78-B, 7y79-B, 7y8u-E, 7y9b-A, 7ycv-A, 7ymo-A, 7yrt-C, 7yta-B, 7yvt-B,
7yvz-A, 7ywq-A, 7z06-A, 7zc8-A, 7zgi-B, 7zgm-A, 7zk1-A, 7zty-A, 7zva-A,
7zw9-A, 8a28-A, 8a4a-A, 8ag9-A, 8ajp-A, 8b26-A, 8b55-A, 8b5t-A, 8b5v-A,
8b73-A, 8cwp-A, 8cxl-A, 8d03-A, 8d08-D, 8d7f-A, 8day-A, 8dgg-A, 8di0-C,
8di1-A, 8dkr-B, 8doa-A, 8ds5-A, 8dt0-A, 8dt6-C, 8dte-A, 8dys-A, 8e8t-B,
8e8u-C, 8gxf-B, 8qcw-A
```

List of 70 targets in CASP14 (Pereira et al., 2021):

```
T1024, T1025, T1026, T1027, T1028, T1029, T1030, T1031, T1032, T1033,
T1034, T1035, T1036s1, T1037, T1038, T1039, T1040, T1041, T1042, T1043,
T1045s1, T1045s2, T1046s1, T1046s2, T1047s1, T1047s2, T1048, T1049, T1050,
T1052, T1053, T1054, T1055, T1056, T1057, T1058, T1060s2, T1060s3, T1061,
T1062, T1064, T1065s1, T1065s2, T1067, T1068, T1070, T1072s1, T1073, T1074,
T1076, T1078, T1079, T1080, T1082, T1083, T1084, T1087, T1088, T1089,
T1090, T1091, T1092, T1093, T1094, T1095, T1096, T1098, T1099, T1100,
T1101
```

### D.5 EVALUATION PIPELINE

**Folding.** In evaluation for folding tasks (Tab. 9), all metrics for all-atom models are computed using OpenStructure (Biasini et al., 2013) unless mentioned otherwise. In particular, we deploy the official docker image of OpenStructure 2.9.1[3] and use the following command to evaluate the structures.

```
ost compare-structures \
-m {MODEL_FILE} \
-r {REFERENCE_FILE} \
-o {OUTPUT_FILE} \
--fault-tolerant --min-pep-length 4 \
--lddt --bb-lddt --rigid-scores --tm-score
```

Notably, for protein folding / generation models that cannot output all-atom structures, we instead adopt the TM-score (Zhang & Skolnick, 2004) for evaluation because the OpenStructure pipeline fails in those cases. We compile the TMscore.cpp c++ source code and compare two structures as follows:

```
TMscore -seq {MODEL_FILE} {REFERENCE_FILE}
```

**MD ensemble generation.** For ATLAS MD ensemble generation (Tab. 2), we base our evaluation pipeline on the dataset split and benchmarking metrics used in previous studies (Jing et al., 2024a;b; Lu et al., 2025), which cover from predicting flexibility to ensemble observables. To obtain the

---

[3]https://git.scicore.unibas.ch/schwede/openstructure/

predicted ensemble, $N = 250$ (Jing et al., 2024a) conformations are sampled from baselines and SimpleFold for each of the 82 test targets, where the median across all targets is reported for each metric. In specific, we report the Pearson's correlation $r$ for pairwise RMSD, global and per-target RMSF; the root mean of 2-Wasserstein distance (W2 distance) and its translation and variance contribution, W2 distance between predicted and true ensembles regarding the first two principal components from PCA by either MD or joint (MD and predicted), and the percentage of samples with cosine similarity > 0.5 between the top principal components of predicted and true ensemble; for the observables, we evaluate the Jaccard similarity (J) of the weak contacts, transient contacts, and exposed residue as well as the Spearman correlation $\rho$ of the exposed mutual information (MI) matrix. We refer the readers to Jing et al. (2024a) for more detailed definition of these metrics. For ESMDiff (Lu et al., 2024a), the Jaccard similarity of exposed residue and the Spearman correlation of exposed MI are left empty because it only generates backbone conformation.

**Two-state prediction.** In order to evaluate the two-state conformation prediction tasks (Tab. 3), we follow the evaluation pipeline in EigenFold (Jing et al., 2023): the global and per-target residue flexbility (in terms of RMSD Pearson's correlation $r$) is calculated after sequence alignment and structural superposition. The TM-ensemble score (at ensemble size 5 following Jing et al. (2023)) is calculated by computing the maximum TM-score (Zhang & Skolnick, 2004) between the ensemble and either ground truth conformation, and averaged across both. We use the same command as above to compute the TMscore.

# E ADDITIONAL EXPERIMENTS

## E.1 DE NOVO AND ORPHAN PROTEINS

We further compare our model with AlphaFold2 and ESMFold on two additional datasets: de novo (designed) proteins and orphan proteins as established in Chowdhury et al. (2022). These evaluation sets are really important because they represent new protein-coding innovations that cannot be traced to ancestral genes (for example proteins that are designed from scratch or those who might evolve so rapidly that they lose detectable homology). The orphan proteins dataset contain 77 targets that have no known sequence homologs (i.e., maximal MSA depth is 1). De novo proteins contain synthetic proteins that were originally de novo designed with computational tools like Rosetta and Amber. We filter it to 62 targets by cutoff data of May-01-2020 such that targets are not used in training all three models. As shown in the following two tables, SimpleFold shows better performance than AlphaFold2 and ESMFold on de novo benchmark. On the orphan protein dataset, SimpleFold shows significant better LDDT than AlplaFold2 while being comparable in other metrics. This evidence supports SimpleFold being a strong generalizable single-sequence folding model that doesn't rely on MSA.

Table 9: Performance of protein folding on the de novo and orphan protein targets.

| Model | TM-score ↑ | GDT-TS ↑ | LDDT ↑ | LDDT-$C_\alpha$ ↑ | RMSD ↓ |
|---|---|---|---|---|---|
| *De Novo* | | | | | |
| AlphaFold2 (Jumper et al., 2021) | 0.831 / 0.866 | 0.850 / 0.898 | 0.781 / 0.805 | 0.876 / 0.894 | 2.950 / 2.307 |
| ESMFold (Lin et al., 2023b) | 0.839 / 0.871 | 0.852 / 0.885 | 0.781 / 0.810 | 0.878 / 0.904 | 3.024 / 1.924 |
| SimpleFold-3B (ours) | 0.852 / 0.880 | 0.877 / 0.928 | 0.807 / 0.823 | 0.906 / 0.922 | 2.729 / 1.535 |
| *De Novo* | | | | | |
| AlphaFold2 (Jumper et al., 2021) | 0.430 / 0.379 | 0.747 / 0.752 | 0.618 / 0.611 | 0.778 / 0.816 | 3.251 / 2.935 |
| ESMFold (Lin et al., 2023b) | 0.391 / 0.320 | 0.700 / 0.706 | 0.485 / 0.471 | 0.731 / 0.761 | 3.775 / 3.329 |
| SimpleFold-3B (ours) | 0.433 / 0.390 | 0.728 / 0.750 | 0.651 / 0.687 | 0.764 / 0.799 | 3.646 / 3.113 |

## E.2 TRAINING WITH SELF-DISTILLED DATA

A relatively common conception for SimpleFold is that such a general purpose architecture and training recipe is only useful as *student* that distills knowledge from *teachers* using strong domain-specific inductive biases (i.e., AlphaFold2 and ESMFold). In practical terms, the concern is that the

general purpose recipe of SimpleFold will completely fail when is not trained on data distilled from other models with strong domain-specific architectures and training objectives (i.e., AlphaFold2 and ESMFold predictions in AFDB and AFESM). In order to clearly understand if this is actually a valid concern we trained SimpleFold models via self-distillation, without using any data from AFDB or AFESM.

We start by training a SimpleFold-700M model on PDB data only (SimpleFold-700M-PDB in Tab. 10). We then use SimpleFold-700M-PDB to generate self-distillation data (on the same protein sequences contained in the filtered SwissProt subset of AFDB and AFESM datasets for a fair comparison) and train a new model, SimpleFold-700M-R1, on this self-distilled data. Finally, we perform a second step of self-distillation where we take SimpleFold-700M-R1 and use it to generate self-distillation data one more time and train a final model which we denote as SimpleFold-700M-R2. All these models follow the training paradigm described in Sect. 4.1 of the main paper. Structures with pLDDT larger than 80 are included in the pre-training phase and those with pLDDT larger than 85 are included in the finetuning phase. It is noted that we train a separate pLDDT modules for both self-distilled version of SimpleFold-700M on PDB data only, and we use these pLDDT modules to filter the self-distilled data.

Our results on both CASP14 and CAMEO22 on Tab. 10 show that SimpleFold does not necessarily require training data distilled from other models to obtain reasonable performance. While training on AFDB/AFESM data provides an edge (potentially due to the use of MSA in AlphaFold2), it does not represent a fundamental requirement for SimpleFold.

Table 10: Performance of SimpleFold trained with self-distilled data.

| Model | TM-score ↑ | GDT-TS ↑ | LDDT ↑ | LDDT-$C_\alpha$ ↑ | RMSD ↓ |
|---|---|---|---|---|---|
| *CAMEO22* | | | | | |
| SimpleFold-700M-PDB | 0.785 / 0.864 | 0.726 / 0.767 | 0.703 / 0.719 | 0.799 / 0.826 | 5.565 / 3.240 |
| SimpleFold-700M-R1 | 0.798 / 0.876 | 0.741 / 0.802 | 0.725 / 0.756 | 0.813 / 0.847 | 5.435 / 3.158 |
| SimpleFold-700M-R2 | 0.805 / 0.878 | 0.749 / 0.796 | 0.727 / 0.754 | 0.819 / 0.858 | 5.160 / 3.106 |
| SimpleFold-700M-AFDB/AFESM | 0.829 / 0.915 | 0.788 / 0.845 | 0.775 / 0.809 | 0.850 / 0.886 | 4.557 / 2.423 |
| *CASP14* | | | | | |
| SimpleFold-700M-PDB | 0.606 / 0.573 | 0.501 / 0.507 | 0.555 / 0.586 | 0.644 / 0.671 | 12.796 / 9.504 |
| SimpleFold-700M-R1 | 0.644 / 0.695 | 0.556 / 0.577 | 0.604 / 0.636 | 0.694 / 0.760 | 11.482 / 7.581 |
| SimpleFold-700M-R2 | 0.649 / 0.698 | 0.565 / 0.595 | 0.601 / 0.621 | 0.696 / 0.765 | 11.327 / 6.605 |
| SimpleFold-700M-AFDB/AFESM | 0.680 / 0.767 | 0.591 / 0.668 | 0.630 / 0.674 | 0.714 / 0.763 | 9.289 / 4.431 |

### E.3 CONFIDENCE MEASURE WITH PLDDT

Fig. 8(a) shows an example of a predicted structure with pLDDT where red and orange denotes low pLDDT and blue denotes high pLDDT. As illustrated, SimpleFold is confident about most predictions of secondary structures while being uncertain about flexible loops. Fig. 8(b) and (c) depict comparison of pLDDT and actual LDDT-$C_\alpha$. We include targets from CAMEO22 and 1000 random selected protein chains from PDB after Jan 2023. pLDDT achieves the Pearson's corelation of 0.77 w.r.t LDDT-$C_\alpha$, which indicates that pLDDT module of SimpleFold correctly models the overall quality of predicted structures. It is also noted that our pLDDT module does not adhere to the generative flow process to output pLDDT. Therefore, it can be applied to measure the quality of predictions from other models seamlessly, which we leave for future investigation.

### E.4 MD ENSEMBLE GENERATION

Tab. 11 lists the results of SimpleFold and SimpleFold-MD on MD ensemble generation of AT-LAS. In particular, no tuning is applied to SimpleFold whereas SimpleFold-MD is tuned on ATLAS training data. It is shown that on MD ensemble generation, SimpleFold also benefits from scaling, namely, larger SimpleFold and SimpleFold-MD achieve better performance.

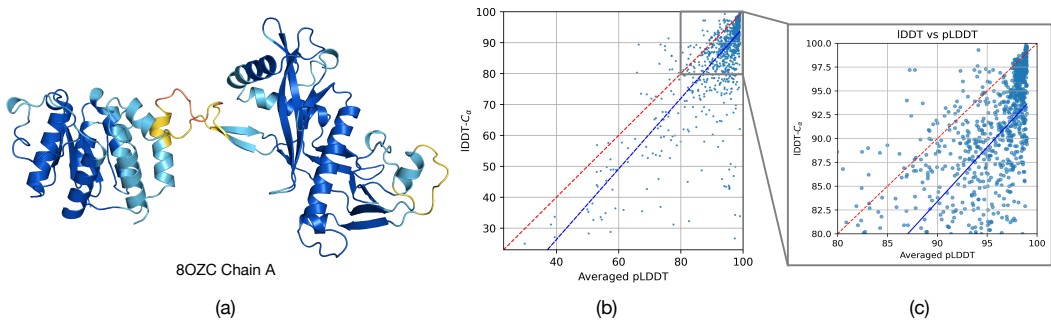

Figure 8: (a) An example prediction of SimpleFold with pLDDT (color red to dark blue denote pLDDT low to high following visualization from Chakravarty & Porter (2022)). (b) & (c) Comparison of pLDDT and LDDT-$C_\alpha$.

Table 11: Evaluation of SimpleFold (SF) of different sizes on MD ensembles.

| | *No Tuning* | | | | | | *Tuned* | | | | | |
| | SF-100M | SF-360M | SF-700M | SF-1.1B | SF-1.6B | SF-3B | SF-MD-100M | SF-MD-360M | SF-MD-700M | SF-MD-1.1B | SF-MD-1.6B | SF-MD-3B |
|---|---|---|---|---|---|---|---|---|---|---|---|---|
| **Pairwise RMSD r** ↑ | 0.17 | 0.21 | 0.29 | 0.30 | 0.38 | 0.44 | 0.19 | 0.27 | 0.30 | 0.32 | 0.40 | 0.45 |
| **Global RMSF r** ↑ | 0.23 | 0.27 | 0.33 | 0.36 | 0.42 | 0.45 | 0.26 | 0.34 | 0.38 | 0.39 | 0.45 | 0.48 |
| **Per target RMSF r** ↑ | 0.59 | 0.63 | 0.65 | 0.64 | 0.63 | 0.60 | 0.62 | 0.67 | 0.67 | 0.68 | 0.68 | 0.67 |
| **RMWD** ↓ | 5.41 | 4.36 | 4.35 | 4.26 | 4.20 | 4.22 | 5.88 | 4.71 | 4.56 | 4.12 | 4.07 | 4.17 |
| **RMWD trans contri** ↓ | 4.83 | 4.02 | 3.95 | 3.84 | 3.79 | 3.74 | 5.32 | 4.22 | 4.19 | 3.60 | 3.44 | 3.40 |
| **RMWD var contri** ↓ | 2.24 | 1.76 | 1.69 | 1.68 | 1.74 | 1.75 | 2.21 | 1.91 | 1.80 | 1.79 | 1.78 | 1.88 |
| **MD PCA W2** ↓ | 1.79 | 1.54 | 1.43 | 1.58 | 1.57 | 1.62 | 1.86 | 1.34 | 1.51 | 1.39 | 1.37 | 1.34 |
| **Joint PCA W2** ↓ | 4.49 | 2.89 | 2.82 | 2.91 | 2.65 | 2.59 | 4.78 | 3.36 | 3.37 | 2.85 | 2.29 | 2.18 |
| **% PC sim > 0.5** ↑ | 30 | 29 | 28 | 32 | 34 | 37 | 28 | 30 | 37 | 37 | 37 | 38 |
| **Weak contacts J** ↑ | 0.47 | 0.43 | 0.43 | 0.44 | 0.36 | 0.36 | 0.52 | 0.55 | 0.57 | 0.58 | 0.58 | 0.56 |
| **Transient contacts J** ↑ | 0.25 | 0.30 | 0.31 | 0.30 | 0.28 | 0.27 | 0.25 | 0.32 | 0.33 | 0.35 | 0.36 | 0.34 |
| **Exposed residue J** ↑ | 0.47 | 0.48 | 0.46 | 0.50 | 0.41 | 0.39 | 0.55 | 0.62 | 0.60 | 0.62 | 0.63 | 0.60 |
| **Exposed MI matrix** $\rho$ ↑ | 0.24 | 0.23 | 0.24 | 0.24 | 0.16 | 0.14 | 0.29 | 0.31 | 0.33 | 0.35 | 0.33 | 0.32 |

## E.5 LDDT Loss

LDDT loss plays an important role in SimpleFold training. In practice, we find LDDT loss is required to generate structures with refined local atomic positions, which largely affects the LDDT metric in folding tasks. We also find that in the second training phase when finetuning the pretrained model on high-quality data, PDB and SwissProt (filtered at pLDDT > 85). Adding a loss weight $\alpha = 1 + 8\text{ReLU}(t - 0.5)$ (Eq. 3) helps getting better results than keeping $\alpha = 1$ as pretraining. Tab. 12 shows the effect of different LDDT loss weighting strategies in finetuning. Applying loss weight schedule $1 + 8 * \text{ReLU}(t - 0.5)$ achieves best overall performance.

Table 12: Ablation of LDDT loss weighting on CAMEO22.

| Model | $\alpha(t)$ | TM-score ↑ | GDT-TS ↑ | LDDT ↑ | LDDT-$C_\alpha$ ↑ | RMSD ↓ |
|---|---|---|---|---|---|---|
| SimpleFold-700M | 0.0 | 0.831 / 0.907 | 0.785 / 0.845 | 0.711 / 0.746 | 0.847 / 0.882 | 4.445 / 2.423 |
| SimpleFold-700M | 1.0 | 0.831 / 0.913 | 0.785 / 0.844 | 0.767 / 0.797 | 0.846 / 0.884 | 4.586 / 2.742 |
| SimpleFold-700M | $1 + 8 * \text{ReLU}(t - 0.5)$ | 0.826 / 0.904 | 0.784 / 0.844 | 0.762 / 0.788 | 0.848 / 0.884 | 4.476 / 2.588 |

## E.6 Inference Settings

Tab. 13, Tab. 14, and Tab. 15 show the ablation of inference settings of SimpleFold-700M on CAMEO22, CASP14, and Apo/Fold-switch, respectively. By default, we set number of steps to 500, $\tau = 0.01$, and $w(t) = \frac{1-t}{t}$ for folding tasks while set $\tau = 0.8$ for multi-state tasks to encourage stochasticity in inference.

Table 13: Ablation of inference settings on CAMEO22.

| # Steps | $\tau$ | $w(t)$ | log time | TM-score ↑ | GDT-TS ↑ | LDDT ↑ | LDDT-$C_\alpha$ ↑ | RMSD ↓ |
|---|---|---|---|---|---|---|---|---|
| 500 | 0.0 | $\frac{1-t}{t}$ | T | 0.831 / 0.918 | 0.790 / 0.844 | 0.777 / 0.815 | 0.851 / 0.887 | 4.497 / 2.410 |
| 500 | 0.01 | $\frac{1-t}{t}$ | T | 0.829 / 0.915 | 0.788 / 0.845 | 0.775 / 0.809 | 0.850 / 0.886 | 4.557 / 2.423 |
| 500 | 0.02 | $\frac{1-t}{t}$ | T | 0.831 / 0.919 | 0.788 / 0.845 | 0.775 / 0.808 | 0.850 / 0.886 | 4.501 / 2.519 |
| 500 | 0.05 | $\frac{1-t}{t}$ | T | 0.831 / 0.913 | 0.787 / 0.848 | 0.775 / 0.808 | 0.849 / 0.885 | 4.461 / 2.429 |
| 500 | 0.1 | $\frac{1-t}{t}$ | T | 0.830 / 0.913 | 0.785 / 0.839 | 0.773 / 0.807 | 0.848 / 0.884 | 4.574 / 2.452 |
| 500 | 0.2 | $\frac{1-t}{t}$ | T | 0.826 / 0.909 | 0.781 / 0.832 | 0.768 / 0.805 | 0.845 / 0.882 | 4.597 / 2.558 |
| 500 | 0.01 | $\tan\left(\frac{\pi(1-t)}{2}\right)$ | T | 0.833 / 0.917 | 0.788 / 0.845 | 0.775 / 0.803 | 0.848 / 0.884 | 4.504 / 2.403 |
| 500 | 0.01 | $\frac{1}{t}$ | T | 0.820 / 0.904 | 0.768 / 0.818 | 0.005 / 0.001 | 0.826 / 0.857 | 4.665 / 2.443 |
| 500 | 0.01 | $\frac{1-t^2}{t}$ | T | 0.829 / 0.916 | 0.788 / 0.841 | 0.775 / 0.808 | 0.849 / 0.885 | 4.571 / 2.421 |
| 500 | 0.01 | $\frac{1-t}{t}$ | F | 0.832 / 0.919 | 0.790 / 0.848 | 0.776 / 0.811 | 0.851 / 0.886 | 4.473 / 2.427 |
| 250 | 0.01 | $\frac{1-t}{t}$ | T | 0.828 / 0.916 | 0.785 / 0.850 | 0.776 / 0.808 | 0.849 / 0.884 | 4.626 / 2.445 |
| 200 | 0.01 | $\frac{1-t}{t}$ | T | 0.831 / 0.915 | 0.788 / 0.843 | 0.777 / 0.808 | 0.850 / 0.888 | 4.417 / 2.491 |
| 150 | 0.01 | $\frac{1-t}{t}$ | T | 0.826 / 0.912 | 0.785 / 0.845 | 0.774 / 0.806 | 0.850 / 0.885 | 4.581 / 2.478 |
| 100 | 0.01 | $\frac{1-t}{t}$ | T | 0.821 / 0.902 | 0.779 / 0.839 | 0.769 / 0.802 | 0.847 / 0.884 | 4.922 / 2.450 |
| 50 | 0.01 | $\frac{1-t}{t}$ | T | 0.654 / 0.618 | 0.605 / 0.599 | 0.615 / 0.641 | 0.717 / 0.746 | 10.971 / 11.024 |

Table 14: Ablation of inference settings on CASP14.

| # Steps | $\tau$ | $w(t)$ | log time | TM-score ↑ | GDT-TS ↑ | LDDT ↑ | LDDT-$C_\alpha$ ↑ | RMSD ↓ |
|---|---|---|---|---|---|---|---|---|
| 500 | 0.0 | $\frac{1-t}{t}$ | T | 0.684 / 0.762 | 0.591 / 0.678 | 0.628 / 0.662 | 0.713 / 0.762 | 9.184 / 4.226 |
| 500 | 0.01 | $\frac{1-t}{t}$ | T | 0.680 / 0.767 | 0.591 / 0.668 | 0.630 / 0.674 | 0.714 / 0.763 | 9.289 / 4.431 |
| 500 | 0.02 | $\frac{1-t}{t}$ | T | 0.677 / 0.770 | 0.589 / 0.667 | 0.629 / 0.676 | 0.712 / 0.758 | 9.319 / 4.645 |
| 500 | 0.05 | $\frac{1-t}{t}$ | T | 0.675 / 0.778 | 0.587 / 0.665 | 0.624 / 0.661 | 0.711 / 0.767 | 9.521 / 4.867 |
| 500 | 0.1 | $\frac{1-t}{t}$ | T | 0.675 / 0.779 | 0.585 / 0.668 | 0.621 / 0.662 | 0.706 / 0.766 | 9.391 / 5.029 |
| 500 | 0.01 | $\frac{1-t}{t}$ | T | 0.673 / 0.780 | 0.585 / 0.647 | 0.617 / 0.652 | 0.708 / 0.764 | 9.167 / 5.018 |
| 500 | 0.01 | $\tan\left(\frac{\pi(1-t)}{2}\right)$ | T | 0.683 / 0.768 | 0.591 / 0.660 | 0.629 / 0.638 | 0.714 / 0.753 | 8.787 / 4.294 |
| 500 | 0.01 | $\frac{1}{t}$ | T | 0.671 / 0.737 | 0.572 / 0.642 | 0.005 / 0.002 | 0.691 / 0.742 | 9.414 / 4.876 |
| 500 | 0.01 | $\frac{1-t^2}{t}$ | T | 0.680 / 0.767 | 0.590 / 0.669 | 0.626 / 0.668 | 0.712 / 0.756 | 9.313 / 4.388 |
| 500 | 0.01 | $\frac{1-t}{t}$ | F | 0.677 / 0.763 | 0.586 / 0.671 | 0.626 / 0.656 | 0.709 / 0.758 | 9.317 / 4.281 |
| 250 | 0.01 | $\frac{1-t}{t}$ | T | 0.679 / 0.777 | 0.593 / 0.683 | 0.627 / 0.677 | 0.715 / 0.764 | 9.374 / 4.765 |
| 200 | 0.01 | $\frac{1-t}{t}$ | T | 0.677 / 0.767 | 0.588 / 0.699 | 0.624 / 0.660 | 0.713 / 0.758 | 9.363 / 4.544 |
| 150 | 0.01 | $\frac{1-t}{t}$ | T | 0.657 / 0.742 | 0.572 / 0.624 | 0.625 / 0.641 | 0.709 / 0.750 | 11.305 / 7.234 |
| 100 | 0.01 | $\frac{1-t}{t}$ | T | 0.633 / 0.680 | 0.558 / 0.584 | 0.615 / 0.657 | 0.701 / 0.759 | 12.370 / 6.615 |
| 50 | 0.01 | $\frac{1-t}{t}$ | T | 0.481 / 0.358 | 0.402 / 0.305 | 0.452 / 0.374 | 0.554 / 0.465 | 17.792 / 17.868 |

Table 15: Ablation of inference settings on Apo and Fold-switch.

| $\tau$ | Res. flex. (global) ↑ | Res. flex. (per-target) ↑ | TM-ens ↑ | Res. flex. (global) ↑ | Res. flex. (per-target) ↑ | TM-ens ↑ |
|---|---|---|---|---|---|---|
| | | Apo/holo | | | Fold-switch | |
| 0.2 | 0.466 | 0.484 / 0.478 | 0.868 / 0.901 | 0.297 | 0.281 / 0.245 | 0.699 / 0.750 |
| 0.4 | 0.538 | 0.501 / 0.512 | 0.869 / 0.901 | 0.314 | 0.305 / 0.228 | 0.697 / 0.748 |
| 0.6 | 0.531 | 0.513 / 0.510 | 0.870 / 0.901 | 0.302 | 0.313 / 0.289 | 0.695 / 0.734 |
| 0.8 | 0.552 | 0.524 / 0.538 | 0.870 / 0.899 | 0.307 | 0.328 / 0.310 | 0.693 / 0.713 |
| 1.0 | 0.562 | 0.525 / 0.520 | 0.867 / 0.896 | 0.319 | 0.337 / 0.329 | 0.687 / 0.716 |

# F  ADDITIONAL VISUALIZATION

## F.1  FOLDING

## F.2  ENSEMBLE GENERATION

## F.3  FAILURE CASES

Fig. 11 shows some examples of failure cases from CAMEO22 and CASP14. In particular, we show predictions with TM-score smaller than 0.6 and also include predictions from ESMFold (Lin et al., 2023b). In these shown cases, SimpleFold mostly predicts the secondary structures correctly. However, the relative positions between the different secondary structure domains are not well modeled.

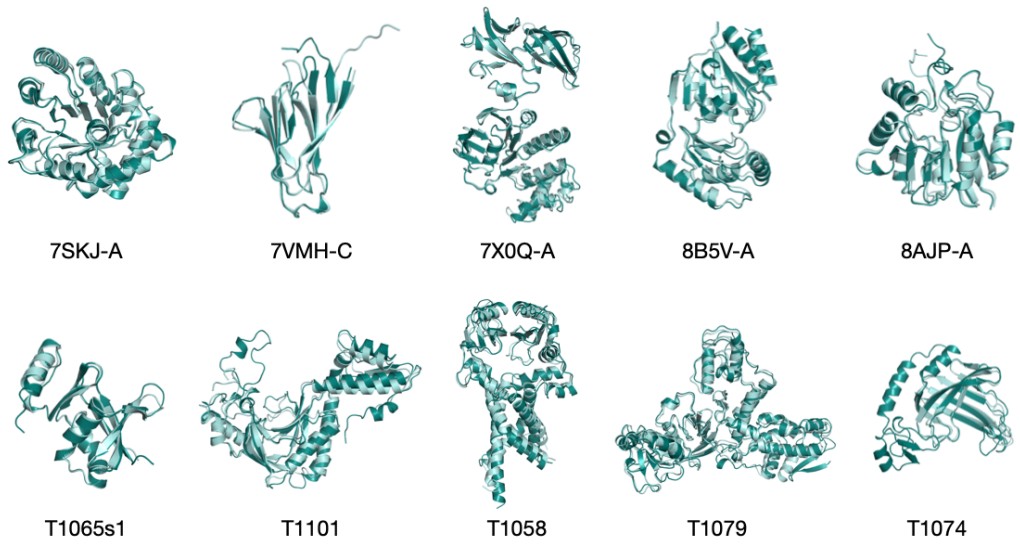

Figure 9: Examples of folding results from SimpleFold with ground truth shown in light aqua and prediction in deep tea (first row from CAMEO22 targets and second row from CASP14 targets).

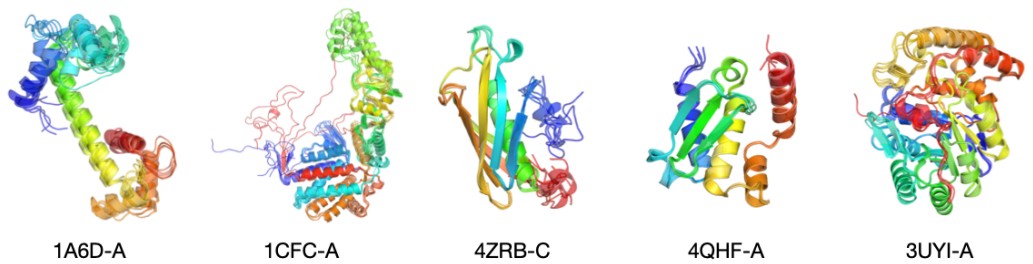

Figure 10: Examples of ensemble generation results from SimpleFold. We align 5 generated conformations of the same protein for visulization.

Interestingly, this failure mode can also be observed in ESMFold, e.g., 7SZ2-B and 7WF9-A. We attribute this to the ESM2 embedding shared by SimpleFold and ESMFold. This indicates a future direction to build more powerful protein language models for representation learning that further benefits protein folding models.

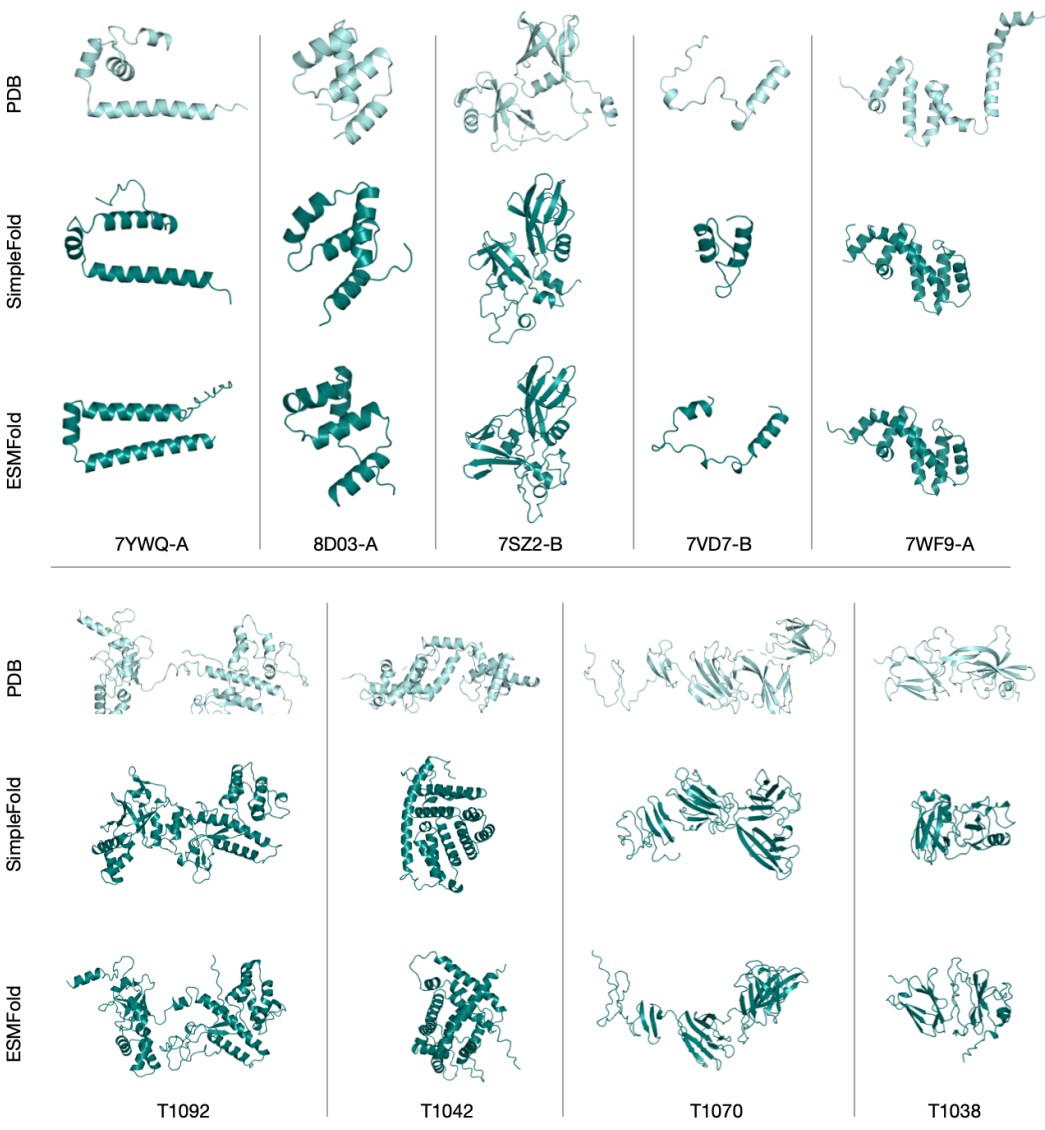

Figure 11: Examples of failure cases (TM-score < 0.6) of SimpleFold predictions with ground truth shown in light aqua and prediction in deep tea (first row from CAMEO22 targets and second row from CASP14 targets). We also include predictions from ESMFold for comparison.

