# OpenReview forum: "SimpleFold: Folding Proteins is Simpler than You Think"
_ICLR.cc/2026/Conference — ICLR 2026 Poster_

### Official Review · Reviewer_WgMN · 2025-10-31

**Soundness:** 3
**Presentation:** 3
**Contribution:** 2
**Rating:** 4
**Confidence:** 5

**Summary:**

This paper introduces SimpleFold, a novel protein folding model based on flow-matching and a standard Transformer architecture. The central thesis challenges the necessity of complex, domain-specific architectural components (like MSAs, explicit pair representations, triangle updates, equivariant modules) prevalent in state-of-the-art models such as AlphaFold2. SimpleFold adopts a generative approach, treating folding as analogous to text-to-image generation, mapping amino acid sequences (encoded via a frozen PLM, ESM2-3B ) to all-atom 3D coordinates using only standard Transformer blocks with adaptive layers conditioned on time. The authors scale SimpleFold up to 3 billion parameters and train it on a large dataset combining PDB experimental structures and approximately 9 million distilled structures . On standard benchmarks (CAMEO22, CASP14), SimpleFold-3B achieves performance competitive with SOTA baselines. Notably, it demonstrates strong performance in ensemble prediction tasks, which is often challenging for models trained with deterministic objectives. The work suggests that scale and general-purpose architectures might suffice for learning complex folding patterns, opening alternative design avenues.

**Strengths:**

- The core argument—that domain-specific inductive biases might be replaceable by scale and general architectures—is highly stimulating and challenges conventional wisdom in protein folding. If validated further, this could significantly simplify model design in structural biology and beyond.
- SimpleFold's reliance solely on standard Diffusion Transformer blocks makes the architecture remarkably simple compared to those with attention bias in AF3. This generality facilitates leveraging advances from the broader Transformer ecosystem and potentially simplifies implementation and optimization (like flash-attn and deepspeed).
- The paper provides a clear demonstration of performance scaling with model size (up to 3B parameters) and training compute on folding benchmarks. This empirical evidence supports the claim that scaling is a viable path for improving performance with this simpler architecture. The scaling with data size is also shown.

**Weaknesses:**

- While competitive, SimpleFold-3B does not consistently surpass the very best models like AlphaFold2, especially on the challenging CASP14 benchmark. This suggests that current domain-specific designs and/or MSA information still provide an edge, particularly for difficult targets.
- SimpleFold 1.6B and 3B models achieve accuracy comparable to (or on par with) ESMFold. However, their inference speed is noted to be slower. This presents a critical trade-off, and the justification for using these models over ESMFold is unclear given this substantial speed limitation.

**Questions:**

- How frequently do SimpleFold's predictions contain steric clashes or distorted covalent geometries compared to models like AlphaFold2 (before relaxation) or other generative models? Is a relaxation step necessary/used for evaluation?
- Given the performance gap on CASP14, do the authors believe further scaling (model size, data) can close this gap, or are there fundamental limitations to the purely data-driven approach without explicit biases for certain protein families or topologies?

---

> ### Author Response · Authors · 2025-11-18
> **Official Comment to Reviewer WgMN**
>
> * Domain-specific designs and/or MSA information still provide an edge, particularly for difficult targets.
>     * We agree with the reviewer that AlphaFold2, shows strong overall performance in CASP14. However, there’s a couple of points that are worth noting:
>         * We want to highlight that it is a fully open question if what makes AlphaFold2 work well in CASP14 are the architectural design or the fact that it uses additional information in the form of close homologs to the target protein sequence. **It is possible that the performance improvement of AlphaFold2 over SimpleFold can be solely explained by the fact that close homologs have additional information and not by the specific way this information gets processed in the network**. To support this claim we refer the reviewer the fact that ESMFold uses the structure model of AlphaFold2 (including pair representations and triangle attention) and yet SimpleFold shows better performance than ESMFold on CASP14, with a general purpose architecture for structure prediction. Extending SimpleFold to have access to homologs of the target protein sequence while keeping a general purpose architecture falls out of the scope of this paper but we think is a great avenue of progress for future work.
>
>         * In addition, MSA might not always be available, for example for de novo or orphan proteins. To clearly showcase the performance of SimpleFold in this setting, we compare our model with AlphaFold2 and ESMFold on two additional datasets consisting of both **de novo (designed) proteins** and **orphan proteins** as established in [1]. These evaluation sets are really important because they represent new protein-coding innovations that cannot be traced to ancestral genes (for example proteins that might evolve so rapidly that they lose detectable homology).  The orphan proteins dataset contain 77 targets that have no known sequence homologs (maximal MSA depth = 1). De novo proteins contain synthetic proteins that were originally de novo designed with computational tools like Rosetta and Amber. We filter it to 62 targets by cutoff data of May-01-2020 such that targets are not used in training all three models. As shown in the following two tables, **SimpleFold shows better performance than AlphaFold2 and ESMFold on de novo benchmark**. On the orphan protein dataset, SimpleFold shows significant better LDDT than AlplaFold2 while being comparable in other metrics. This evidence supports SimpleFold being a strong generalizable single-sequence folding model that doesn’t rely on MSA. We have included the results in Appendix E.1 of updated manuscript.
>         * Orphan
>         * | | TM-Score | GDT-TS | LDDT | BB-LDDT | RMSD |
>             |---|---|---|---|---|---|
>             |AlphaFold2 |0.430 / 0.379|0.747 / 0.752|0.618 / 0.611|0.778 / 0.816|3.251 / 2.935|
>             |ESMFold    |0.391 / 0.320|0.700 / 0.706|0.485 / 0.471|0.731 / 0.761|3.775 / 3.329|
>             |SimpleFold |0.433 / 0.390|0.728 / 0.750|0.651 / 0.687|0.764 / 0.799|3.646 / 3.113|
>         * De Novo
>         * | | TM-Score | GDT-TS | LDDT | BB-LDDT | RMSD |
>             |---|---|---|---|---|---|
>             |AlphaFold2|0.831 / 0.866|0.850 / 0.898|0.781 / 0.805|0.876 / 0.894|2.950 / 2.307|
>             |ESMFold   |0.839 / 0.871|0.852 / 0.885|0.781 / 0.810|0.878 / 0.904|3.024 / 1.924|
>             |SimpleFold|0.852 / 0.880|0.877 / 0.928|0.807 / 0.823|0.906 / 0.922|2.729 / 1.535|
>
>
> * Inference speed
>     * SimpleFold is a flow-matching generative model that’s no different than other text-to-image or text-to-3D generative models. As a result, **it is perfectly possible to adopt established distillation methods for these models to decrease the number of timesteps required at inference time, like consistency models [2] and/or meanflows [3]**. These and several other approaches have achieved great generation results with less than 5 inference steps. **For example, a SimpleFold-3B model distilled to use 5 timesteps at inference would take less than 1s to generate a structure for a protein sequence with 512 residues while ESMFold takes more than 7s (both on H100 gpus)**. We highlight to the reviewer that the major goal of this work is to demonstrate the effectiveness of a protein folding model using a general purpose architecture and trained with a generative objective and not optimizing inference time of such models, which we believe can be done in future work.
>     * In addition SimpleFold offers other benefits over ESMFold. **For example, it achieves very strong performance on multi-state conformation prediction tasks like apo/holo and fold-switch. Whereas ESMFold, which is trained with a deterministic regression objective, fails to generative different conformations for one given target.** Furthermore, in both multi-state conformation prediction and folding benchmarks, SimpleFold surpasses ESMFlow by large margins, which is a finetuned model from ESMFold with a flow-matching generative objective.

---

> > ### Author Response · Authors · 2025-11-18
> > **Official Comment to Reviewer WgMN (2)**
> >
> > * Physically infeasible geometries and necessity of relaxation step.
> >     * **In evaluation, we very rarely find SimpleFold generating physically infeasible structures like steric clashes or distorted covalent geometries.** In fact, stereochemical plausibility is assessed in LDDT [4]. Particularly, residues with infeasible chemical bond lengths, close atomic interaction, or impossible bond angles result in lower LDDT scores. SimpleFold shows competitive LDDT scores on standard folding benchmarks like CASP14 and CAMEO22. In particular, the longest target we tested is T1061 from CASP14 with 949 residues, and generated results still obey physical rules though not explicitly forced by architectural designs. **We want to highlight that no relaxation step is needed for evaluation, which makes SimpleFold end-to-end in inference. **
> > * Are there fundamental limitations to purely data-driven approach without explicit biases?
> >     * We thank the reviewer bringing up this valuable question. From our perspective, there is no fundamental limitation to rely on a general-purpose and scalable approach like SimpleFold for protein folding. We believe that by further scaling up model and data size, we can bridge the gap to strong folding models like AlphaFold.
> >
> >         We agree with the reviewer that all learning-based models would face challenges for certain protein topologies / families where limited or even no structure data can be found for training (for example SimpleFold outperforms AlphaFold2 on de novo proteins while AlphafFold2 outperforms SimpleFold on CASP14). However, we believe that the strategy of building a simplified and scalable **generative framework like SimpleFold provides an effective approach** to build a generalizable folding model for the following reasons:
> >     * (1) **Due to its generative nature, SimpleFold can trivially make use of simulation data during training** (like molecular dynamics simulation), where an ensemble of multiple possible conformations is computed for one single protein instead of requiring purely deterministic structures as  in PDB. This can be helpful for protein families that are hard to crystallize where experimental structures are currently lacking.
> >     * (2) While MSA does provide additional evolutionary information which can result in superior performance on certain targets like CASP14, it can be detrimental for targets where MSA is inaccessible (e.g., orphan and de novo proteins). In this setting, SimpleFold shows strong performance compared to AlphaFold2/ESMFold, which demonstrates its generalization to different targets. Moreover, one may not necessarily require sophisticated domain-specific architectures to utilize MSA. For example, recent work [5] has integrated MSA into pretrained pLM, which provides an alternative for current MSA modeling in AlphaFold2.
> >
> > References:
> >
> > [1] Chowdhury, Ratul, et al. "Single-sequence protein structure prediction using a language model and deep learning." Nature Biotechnology 40.11 (2022): 1617-1623.
> >
> > [2] Song, Yang, et al. "Consistency models." (2023).
> >
> > [3] Geng, Zhengyang, et al. "Mean flows for one-step generative modeling." arXiv preprint arXiv:2505.13447 (2025).
> >
> > [4] Mariani, Valerio, et al. "lDDT: a local superposition-free score for comparing protein structures and models using distance difference tests." Bioinformatics 29.21 (2013): 2722-2728.
> >
> > [5] Jain, Sarthak, et al. "E1: Retrieval-Augmented Protein Encoder Models." bioRxiv (2025): 2025-11.

---

> > > ### Comment · Reviewer_WgMN · 2025-11-26
> > >
> > > Thanks for the authors' rebuttal, which address most of my concerns. I've raised my score.

---

> > > > ### Author Response · Authors · 2025-11-26
> > > > **Official Comment to Reviewer WgMN**
> > > >
> > > > We thank the reviewer for acknowledging our work. We're happy to answer any additional questions.

---

> > > ### Author Response · Authors · 2025-11-28
> > >
> > > We’re glad to hear that we addressed most of your concerns, and appreciate that you raised your score to positive (6) on Nov. 25th. Thank you for your feedback!

---

### Official Review · Reviewer_iwqT · 2025-10-31

**Soundness:** 3
**Presentation:** 3
**Contribution:** 2
**Rating:** 4
**Confidence:** 4

**Summary:**

This paper proposes SimpleFold, a flow-matching based protein folding model that relies solely on general-purpose transformer blocks for efficient structure prediction. In contrast to prior methods that employ triangular updates, explicit pair representations, or multi-objective training, SimpleFold utilizes standard transformer blocks enhanced with adaptive layers and is trained with a generative flow-matching objective augmented by a structural loss term. Evaluated on standard folding benchmarks, the SimpleFold-3B model—trained on both distilled structures and PDB data—achieves performance competitive with state-of-the-art baselines. Additionally, the model demonstrates strong capability in ensemble prediction tasks.

**Strengths:**

•	SimpleFold effectively removes complex components commonly used in previous protein folding models—such as MSA processing, pairwise representations, and triangle modules—leading to significant computational speedup while maintaining competitive prediction accuracy.

•	The model shows promising results in ensemble generation, highlighting its potential for capturing conformational diversity.

•	The paper is clearly structured and easy to follow.

**Weaknesses:**

•	The scope of SimpleFold remains limited to single-chain protein structure prediction. Given the emergence of efficient and generalizable predictors such as ESMFold and Protenix-Mini [1]—which are also capable of predicting biomolecular complexes—SimpleFold does not demonstrate a clear advantage in either efficiency or accuracy, raising questions about its practical added value.

•	The overall framework constitutes a relatively straightforward integration of existing techniques from computer vision and protein modeling (e.g., flow matching and transformer adaptations), and the core technical innovation appears incremental.

[1] Protenix-Mini: Efficient Structure Predictor via Compact Architecture, Few-Step Diffusion and Switchable pLM

**Questions:**

•	SimpleFold appears to adopt training structures and strategies similar to prior flow-based folding methods such as AlphaFlow or ESMFlow. Could the authors provide further insight or ablation studies to explain why SimpleFold is more effective at recovering multi-state protein conformations? Is this capability attributable to the architectural design, the training objective, or the data used?

---

> ### Author Response · Authors · 2025-11-18
> **Official Comment to Reviewer iwqT**
>
> * Questions about practical added value by SimpleFold
>     * To clearly showcase the performance of SimpleFold, we compare our model with AlphaFold2 and ESMFold on two additional datasets: **de novo (designed) proteins** and **orphan proteins** as established in [1]. These evaluation sets are really important because they represent new protein-coding innovations that cannot be traced to ancestral genes (for example proteins that might evolve so rapidly that they lose detectable homology).  The orphan proteins dataset contain 77 targets that have no known sequence homologs (maximal MSA depth = 1). De novo proteins contain synthetic proteins that were originally de novo designed with computational tools like Rosetta and Amber. We filter it to 62 targets by cutoff data of May-01-2020 such that targets are not used in training all three models. As shown in the following two tables, **SimpleFold shows better performance than AlphaFold2 and ESMFold on de novo benchmark**. On the orphan protein dataset, SimpleFold shows significant better LDDT than AlplaFold2 while being comparable in other metrics. This evidence supports SimpleFold being a strong generalizable single-sequence folding model that doesn’t rely on MSA. We have included the results in Appendix E.1 of updated manuscript.
>     * Orphan
>     * | | TM-Score | GDT-TS | LDDT | BB-LDDT | RMSD |
>         |---|---|---|---|---|---|
>         |AlphaFold2 |0.430 / 0.379|0.747 / 0.752|0.618 / 0.611|0.778 / 0.816|3.251 / 2.935|
>         |ESMFold    |0.391 / 0.320|0.700 / 0.706|0.485 / 0.471|0.731 / 0.761|3.775 / 3.329|
>         |SimpleFold |0.433 / 0.390|0.728 / 0.750|0.651 / 0.687|0.764 / 0.799|3.646 / 3.113|
>     * De Novo
>     * | | TM-Score | GDT-TS | LDDT | BB-LDDT | RMSD |
>         |---|---|---|---|---|---|
>         |AlphaFold2|0.831 / 0.866|0.850 / 0.898|0.781 / 0.805|0.876 / 0.894|2.950 / 2.307|
>         |ESMFold   |0.839 / 0.871|0.852 / 0.885|0.781 / 0.810|0.878 / 0.904|3.024 / 1.924|
>         |SimpleFold|0.852 / 0.880|0.877 / 0.928|0.807 / 0.823|0.906 / 0.922|2.729 / 1.535|
>     * In addition, we want highlight that SimpleFold achieves very strong performance on multi-state conformation prediction tasks like apo/holo and fold-switch. Whereas ESMFold, which is trained with a deterministic regression objective, fails to generative different conformations for one given target. Note that this is the same SimpleFold checkpoint that achieves comparable performance to ESMFold on standard folding tasks like CASP14 and CAMEO22.
>     * Finally, we agree with reviewer that modeling protein complexes is valuable future work. In this work, we aim to show the viability of **training a folding model using general purpose architectural blocks via a generative objective**, which is a strong departure from previous models. Nonetheless, we plan to explore protein complex modeling in the near future.
>
>
> * Core technical innovation
>     * We agree with the reviewer that SimpleFold does utilize an architecture purely based on standard transformer blocks trained with a generative flow-matching objective to build a folding model. However, we would like to clarify to the reviewer that on one hand, existing approaches like ESMFold which previously used pLMs to replace MSA, do make use of the structure module in AlphaFold2 adopting the **usage of pair representations, triangle attention, etc**.  On the other hand, related approaches like AlphaFlow and ESMFlow use a **generative training objective merely as a fine-tuning strategy** on top of already fully trained checkpoints from AlphaFold2 and ESMFold. Borrowing, as a result, their architectural designs which again include domain specific architectural designs like pair representations, triangle attention, etc.
>     *  **We want to be completely transparent in our goal**: we are not aiming to design a “novel” folding method, but rather asking if we can drop all domain-specific architectural designs (ie. pair representations, triangle attention, etc. ) and **train a protein folding model with a general purpose architecture via a generative objective from scratch**. To the best of our knowledge **this does represent a significant departure from how proteins folding models currently designed** and SimpleFold is the only existing protein folding model trained in this way.

---

> > ### Author Response · Authors · 2025-11-18
> > **Official Comment to Reviewer iwqT (2)**
> >
> > * Effective at recovering multi-state protein conformations
> >     * We thank the reviewer for highlighting our superior performance of SimpleFold on multi-state conformation prediction. Even though **SimpleFold, AlphaFlow and ESMFlow all use a flow-matching training objectives, the architectural design and the training paradigm are drastically different:**
> >     * AlphaFlow and ESMFlow are built upon the AlphaFold and ESMFold network architectures, respectively. This means they inherit domain-specific heuristic architectural designs like pair representation and triangle attention. On the other hand, SimpleFold is based purely on  standard transformer blocks without any domain-specific network blocks. In addition, **AlphaFlow and ESMFlow use a generative training objective merely as a fine-tuning strategy** on top of already fully trained checkpoints from AlphaFold2 and ESMFold which use a deterministic regression objective.  On the contrary, **SimpleFold is built from the ground up to be a pure generative model trained from scratch with a flow-matching objective.**
> >     * Building SimpleFold from the ground up as a generative model that is trained from scratch with a flow-matching objective results in improvements in multi-state benchmarks over models that only fine-tune pre-trained deterministic models like AlphaFold and ESMFold. We have included the discussions in Appendix B.2 of updated manuscript.

---

> ### Author Response · Authors · 2025-11-27
> **Kind request for discussion**
>
> Dear reviewer,
>
> Thank you for your time reviewing our paper and for your suggestions. We made an effort to complete our rebuttal (including new experiments) earlier in the discussion process to have enough time to engage in discussion with you during this period. As we near the end of the discussion period we kindly remind you that we are happy to engage in discussions with you.
>
> Authors

---

### Official Review · Reviewer_MgVM · 2025-11-01

**Soundness:** 2
**Presentation:** 3
**Contribution:** 2
**Rating:** 4
**Confidence:** 4

**Summary:**

The paper presents SimpleFold, a flow-matching generative model for protein folding. It uses a standard transformer architecture to generate full-atom structures, conditioned on embeddings from a frozen ESM2-3B protein language model (PLM). This approach notably avoids the explicit pair representations and triangular updates common in models like AlphaFold. The largest 3B parameter model, trained on approximately 9 million distilled and experimental structures, achieves performance that is competitive with, but not superior to, existing state-of-the-art baselines on standard folding and ensemble generation benchmarks.

**Strengths:**

The primary strength is **architectural simplification**. The work demonstrates that a general-purpose transformer can achieve comparable performance to more complex, bespoke architectures, provided it is leveraged at a large scale and conditioned on a powerful pretrained PLM. The clear empirical validation of scaling laws for both model and data size is a useful, albeit expected, finding.

**Weaknesses:**

**Limited Novelty**: The approach is highly derivative. It combines a standard transformer with a known flow-matching objective and, most importantly, relies heavily on the powerful ESM2-3B PLM. These attempts have already appeared in previous work and are not surprising; they cannot be called new folding methods.

**Insignificant Performance**: In the context of the rapidly advancing protein design field, the results are not a significant breakthrough. The model achieves "competitive" or "comparable" performance, failing to clearly outperform established baselines like AlphaFold2 or ESMFold. This incremental result does not demonstrate a clear advantage.

**Questions:**

See **Weaknesses**.

---

> ### Author Response · Authors · 2025-11-18
> **Official Comment to Reviewer MgVM**
>
> * Novelty:
>     * We agree with the reviewer that SimpleFold does utilize existing pLMs (like ESM2) and an architecture purely based on standard transformer blocks trained with a generative flow-matching objective to build a folding model. However, we would like to clarify to the reviewer that on one hand, existing approaches like ESMFold which previously used pLMs to replace MSA, do make use of the structure module in AlphaFold2 adopting the **usage of pair representations, triangle attention, etc.**  On the other hand, related approaches like AlphaFlow and ESMFlow use a **generative training objective merely as a fine-tuning strategy** on top of already fully trained checkpoints from AlphaFold2 and ESMFold. Borrowing, as a result, their architectural designs which again include domain specific architectural designs like pair representations, triangle attention, etc.
>     * **We want to be completely transparent in our goal**: we are not aiming to design a “novel” folding method, but rather asking if we can drop all domain-specific architectural designs (ie. pair representations, triangle attention, etc. ) and **train a protein folding model with a general purpose architecture via a generative objective from scratch**. To the best of our knowledge **this does represent a significant departure from how proteins folding models currently designed** and SimpleFold is the only existing protein folding model trained in this way.
> * Performance:
>     * To clearly showcase the performance of SimpleFold, we compare our model with AlphaFold2 and ESMFold on two additional datasets: **de novo (designed) proteins** and **orphan proteins** as established in [1]. These evaluation sets are really important because they represent new protein-coding innovations that cannot be traced to ancestral genes (for example proteins that are designed from scratch or those who might evolve so rapidly that they lose detectable homology).  The orphan proteins dataset contain 77 targets that have no known sequence homologs (maximal MSA depth = 1). De novo proteins contain synthetic proteins that were originally de novo designed with computational tools like Rosetta and Amber. We filter it to 62 targets by cutoff data of May-01-2020 such that targets are not used in training all three models. As shown in the following two tables, **SimpleFold shows better performance than AlphaFold2 and ESMFold on de novo benchmark**. On the orphan protein dataset, SimpleFold shows significant better LDDT than AlplaFold2 while being comparable in other metrics. This evidence supports SimpleFold being a strong generalizable single-sequence folding model that doesn’t rely on MSA. We have included the results in Appendix E.1 of updated manuscript.
>     * De novo
>     * | | TM-Score | GDT-TS | LDDT | BB-LDDT | RMSD |
>         |---|---|---|---|---|---|
>         |AlphaFold2|0.831 / 0.866|0.850 / 0.898|0.781 / 0.805|0.876 / 0.894|2.950 / 2.307|
>         |ESMFold   |0.839 / 0.871|0.852 / 0.885|0.781 / 0.810|0.878 / 0.904|3.024 / 1.924|
>         |SimpleFold|0.852 / 0.880|0.877 / 0.928|0.807 / 0.823|0.906 / 0.922|2.729 / 1.535|
>     * Orphan
>     * | | TM-Score | GDT-TS | LDDT | BB-LDDT | RMSD |
>         |---|---|---|---|---|---|
>         |AlphaFold2 |0.430 / 0.379|0.747 / 0.752|0.618 / 0.611|0.778 / 0.816|3.251 / 2.935|
>         |ESMFold.   |0.391 / 0.320|0.700 / 0.706|0.485 / 0.471|0.731 / 0.761|3.775 / 3.329|
>         |SimpleFold |0.433 / 0.390|0.728 / 0.750|0.651 / 0.687|0.764 / 0.799|3.646 / 3.113|
>
>
>     * We want to also highlight that as a generative model, SimpleFold shows strong performance on ensemble generation tasks like multi-state prediction (i.e., apo/holo and fold-switch) and MD simulation (i.e., ATLAS). Whereas folding models like AlphaFold2 and ESMFold lack diversity in predicted structures given their deterministic training objective. This indicates the benefits of building folding models based on generative objectives and shows more versatile applications for SimpleFold.
>
> Reference:
>
> [1] Chowdhury, Ratul, et al. "Single-sequence protein structure prediction using a language model and deep learning." Nature Biotechnology 40.11 (2022): 1617-1623.

---

> > ### Author Response · Authors · 2025-11-27
> > **Kind request for discussion**
> >
> > Dear reviewer,
> >
> > Thank you for your time reviewing our paper and for your suggestions. We made an effort to complete our rebuttal (including new experiments) earlier in the discussion process to have enough time to engage in discussion with you during this period. As we near the end of the discussion period we kindly remind you that we are happy to engage in discussions with you.
> >
> > Authors

---

### Author Response · Authors · 2025-11-18
**Official Comment to All Reviewers**

To all reviewers:

We thank reviewers for all their feedback, which we believe will help clarify the value of SimpleFold. Here we summarize the main updates:

- A new section has been added to the appendix with evaluations on de novo and orphan proteins to show the performance of SimpleFold compared to AlphaFold2 and ESMFold.
- We have answered concerns from reviews regarding novelty and performance of SimpleFold compared with existing approaches.

We are happy to engage in respectful scientific discussions with all reviewers.

Regards,

Authors of Submission 13107

---

### Author Response · Authors · 2025-11-21
**Kind request for discussion**

Dear reviewers, AC and SACs,

We appreciate your time reviewing our submission. Following reviewers suggestions we have updated our submission and ran new experiments that help clarifying the value of SimpleFold.  We have put substantial effort to complete our rebuttal early in the process and we wanted to send a note to kindly remind you that we are happy and eager to engage in discussions.

Authors

---

### Author Response · Authors · 2025-12-02
**Official Comment by Authors**

Dear AC,

We thank the AC for taking the time reviewing our submission. Here, we summarize our core contribution and main updates in response to questions from reviewers:

1. **Novelty**
    - The goal of this work is to understand if domain-specific architectural designs are required in protein folding models (ie. pair representations, triangle attention, etc. ) and **train a protein folding model with a general purpose transformer architecture via a generative objective from scratch**. To the best of our knowledge **this represents a significant departure from how proteins folding models currently designed** and SimpleFold is the only existing protein folding model trained in this way. In particular, SimpleFold eliminates **usage of pair representations, triangle attention, etc.** (e.g., AlphaFold2, ESMFold) and doesn’t treat **generative training objective merely as a fine-tuning strategy** (e.g., AlphaFold and ESMFlow). We believe SimpleFold represents a valuable direction to build powerful protein generative models.

2. **Performance**
    - We added additional experiments on two datasets: **de novo (designed) proteins** and **orphan proteins**. These evaluation sets represent cases where proteins sequences cannot be traced to evolutionary information.  **SimpleFold shows better performance than AlphaFold2 and ESMFold on de novo benchmark**. On the orphan protein dataset, SimpleFold shows significant better LDDT than AlplaFold2 while being comparable in other metrics. This evidence supports SimpleFold being a strong generalizable single-sequence folding model that doesn’t rely on MSA.
    - De novo
        ||TM-Score|GDT-TS|LDDT|BB-LDDT|RMSD|
        |-|-|-|-|-|-|
        |AlphaFold2|0.831 / 0.866|0.850 / 0.898|0.781 / 0.805|0.876 / 0.894|2.950 / 2.307|
        |ESMFold   |0.839 / 0.871|0.852 / 0.885|0.781 / 0.810|0.878 / 0.904|3.024 / 1.924|
        |SimpleFold|0.852 / 0.880|0.877 / 0.928|0.807 / 0.823|0.906 / 0.922|2.729 / 1.535|
    - Orphan
        ||TM-Score|GDT-TS|LDDT|BB-LDDT|RMSD|
        |-|-|-|-|-|-|
        |AlphaFold2|0.430 / 0.379|0.747 / 0.752|0.618 / 0.611|0.778 / 0.816|3.251 / 2.935|
        |ESMFold|0.391 / 0.320|0.700 / 0.706|0.485 / 0.471|0.731 / 0.761|3.775 / 3.329|
        |SimpleFold|0.433 / 0.390|0.728 / 0.750|0.651 / 0.687|0.764 / 0.799|3.646 / 3.113|
     -  We want to also highlight that as a generative model, SimpleFold shows competitive performance on standard folding benchmarks (e.g., CASP14 and CAMEO22) and strong performance on ensemble generation tasks (e.g., multi-state prediction and MD ensemble). This indicates the benefits of building folding models based on generative objectives and shows more versatile applications for SimpleFold.

3. **Self-distillation**
    - A typical concern for SimpleFold is if such a general purpose architecture and training recipe will completely fail when is not trained on data distilled from other models with strong domain-specific architectures and training objectives (ie. AlphaFold2 and ESMFold predictions in AFDB and AFESM).  In order to clearly understand if this is actually a valid concern we trained SimpleFold models via self-distillation, without using any structures from AFDB or AFESM.
   - We start by training a SimpleFold-700M model on PDB data only (SimpleFold-700M-PDB in the table below). We then use SimpleFold-700M-PDB to generate self-distillation data (on the same protein targets contained in the filtered SwissProt of AFDB and AFESM datasets for a fair comparison) and train a new model, SimpleFold-700M-R1, on this self-distilled data. Finally, we perform a second step of self-distillation where we take SimpleFold-700M-R1 and use it to generate self-distillation data one more time and train a final model which we denote as SimpleFold-700M-R2.
   - Our results on both CASP14 and CAMEO22 show that SimpleFold does not require training data distilled from other models to obtain reasonable performance. While training on AFESM data provides a small but noticeable edge, it does not represent a fundamental requirement.
   - CASP14
        ||TM-Score|GDT-TS|LDDT|BB-LDDT|RMSD|
        |-|-|-|-|-|-|
        |SF-700M-PDB|0.606 / 0.573|0.501 / 0.507| 0.555 / 0.586 |0.644 / 0.671|12.796 / 9.504|
        |SF-700M-R1|0.644 / 0.695|0.556 / 0.577|0.604 / 0.636|0.694 / 0.760|11.482 / 7.581|
        |SF-700M-R2|0.649 / 0.698|0.565 / 0.595|0.601 / 0.621|0.696 / 0.765|11.327 / 6.605|
        |SF-700M-AFDB/AFESM| 0.680 / 0.767 |0.591 / 0.668|0.630 / 0.674 |0.714 / 0.763 |9.289 / 4.431|
    - CAMEO22
        ||TM-Score|GDT-TS|LDDT|BB-LDDT|RMSD|
        |-|-|-|-|-|-|
        |SF-700M-PDB|0.785 / 0.864|0.726 / 0.767|0.703 / 0.719|0.799 / 0.826|5.565 / 3.240|
        |SF-700M-R1|0.798 / 0.876|0.741 / 0.802|0.725 / 0.756|0.813 / 0.847|5.435 / 3.158|
        |SF-700M-R2|0.805 / 0.878|0.749 / 0.796|0.727 / 0.754|0.819 / 0.858|5.160 / 3.106|
        |SF-700M-AFDB/AFESM|0.829 / 0.915|0.788 / 0.845|0.775 / 0.809|0.850 / 0.886|4.557 / 2.423|

---

### Meta-Review · Area_Chair_YCNJ · 2026-01-06

**Summary:**

The reviewers were impressed by the simple architecture and the ensemble generation capabilities of SimpleFold. However, they noted a lack of methodological novelty and state of the art results as well as insufficient evidence that architectural innovations such triangle attention and pair representations are truly not required for structure prediction because SimpleFold was trained on predictions from models that use these architectures. The rebuttal thoroughly addresses each of these concerns.

**Reviewer Concerns:**

All the reviewers present a consistent set of concerns.

**1. The paper lacks methodological novelty.**

The rebuttal clarifies that, while transformers and flow matching are not new, the key, novel result here is that most of the architectural innovations introduced by AlphaFold2 and retained by single-sequence structure predictors such as ESMFold are not required for strong performance. This was not a widely-held view in the field up to this point, so in addition to the clarifications in the rebuttal, my opinion is that the reviewer concerns about lack of methodological novelty misunderstood either the state of the field or the paper's primary contribution. This is further underscored by the inherent inconsistency of praising the paper for simplifying the architecture and then in the next paragraph criticizing the architecture for being insufficiently novel.

**2. The paper lacks state of the art structure prediction results.**

During the rebuttal, the authors evaluated SimpleFold and previous models on a set of de novo and orphan proteins. SimpleFold shows better performance than AlphaFold2 and ESMFold on the de novo proteins. On the orphan protein dataset, SimpleFold shows significantly better LDDT than AlplaFold2 while being comparable in other metrics and outperforms ESMFold. Furthermore, when I look at the original results on CAMEO22 and CASP14, SimpleFold outperforms all single-sequence methods on CASP14 while being comparable on CAMEO22. This is especially notable because SimpleFold uses the same sequence embeddings as ESMFold while removing innovations such as triangle attention and pair representations from the folding trunk and using a generative objective, clearly demonstrating that these architectural innovations are not essential for structure prediction and that generative structure prediction can be accomplished without MSAs or pretraining on structure prediction.

**3. It's unfair to claim that a vanilla transformer is sufficient if the performance relies on augmenting the training set with predictions from AlphaFold and ESMFold.**

During the revision, the authors found that self-distillation is sufficient to produce accurate structure predictions.

**Reviewer Scores:**

I believe each reviewer would have gone from 4 -> 8. All major concerns were clearly addressed during the revision.

---

### Decision · Program_Chairs · 2026-01-26

Accept (Poster)